# Beyond Model Ranking: Predictability-Aligned Evaluation for Time Series Forecasting

## Abstract

In the era of increasingly complex AI models for time series forecasting, progress is often measured by marginal improvements on benchmark leaderboards. However, this approach suffers from a fundamental flaw: standard evaluation metrics conflate a model's performance with the data's intrinsic unpredictability. To address this pressing challenge, we introduce a novel, predictability-aligned diagnostic framework grounded in spectral coherence. Our framework makes two primary contributions: the **Spectral Coherence Predictability (SCP)**, a computationally efficient ($O(N \log N)$) and task-aligned score that quantifies the inherent difficulty of a given forecasting instance, and the **Linear Utilization Ratio (LUR)**, a frequency-resolved diagnostic tool that precisely measures how effectively a model exploits the linearly predictable information within the data. We validate our framework's effectiveness and leverage it to reveal two core insights. First, we provide the first systematic evidence of "predictability drift", demonstrating that a task's forecasting difficulty varies sharply over time. Second, our evaluation reveals a key architectural trade-off: complex models are superior for low-predictability data, whereas linear models are highly effective on more predictable tasks. We advocate for a paradigm shift, moving beyond simplistic aggregate scores toward a more insightful, predictability-aware evaluation that fosters fairer model comparisons and a deeper understanding of model behavior. Codes and data are available at `https://anonymous.4open.science/r/TS_Predictability-C8B7`.

## 1 Introduction

Despite the proliferation of ever-more-complex models for time-series forecasting, true progress in the field remains notoriously difficult to measure (Bergmeir, 2024). The community relies on standard metrics, such as Mean Squared Error (MSE) and Mean Absolute Error (MAE), which quantify the magnitude of prediction errors but fail to reveal their origin. These metrics obscure a critical distinction: is a model failing because of its own limitations, or has it hit the fundamental ceiling of what is predictable in the data (Ehrenberg & Bound, 1993)? This ambiguity creates a severe evaluation dilemma, where a sophisticated model on a chaotic series may appear worse than a simple one. It also stalls scientific progress by wasting significant computational resources on problems where gains are impossible. Disentangling model inadequacy from intrinsic data unpredictability is therefore one of the most pressing challenges in the field today (Erkintalo, 2015).

The ideal solution to this dilemma is to first quantify a time series' intrinsic predictability and then evaluate a model's performance relative to this established baseline. However, designing a predictability metric that is fit for the modern deep learning forecasting paradigm presents a series of formidable challenges (Pennekamp et al., 2019). First, such a metric must be task-aligned, meaning its theoretical foundation should cohere with multi-horizon forecasting under a squared-error loss, rather than traditional single-step classification accuracy (Mishra & Palanisamy, 2018). Second, it must be computationally efficient to handle the massive, high-dimensional time series prevalent today (Fiecas et al., 2019). Finally, a single, global predictability score is insufficient; a truly useful tool must be diagnostic, offering insights to reveal where a model succeeds or fails in capturing predictable patterns.

Viewed through the lens of these challenges, existing tools are ill-suited for this purpose. Traditional proxies for predictability, such as entropy-rate estimators and Lempel-Ziv complexity, suffer from a fundamental paradigm mismatch (Aboy et al., 2006). These information-theoretic methods were developed for symbolic dynamics and classification tasks, designed to estimate an upper bound on next-symbol accuracy under a 0-1 loss, not a lower bound on multi-step regression error (Zhao et al., 2021). Furthermore, they often assume stationarity and are computationally expensive, which typically entails quadratic-to-cubic time complexity, rendering them impractical for the large-scale, non-stationary datasets common in modern applications (Kontoyiannis et al., 2002; Wyner & Ziv, 2002). An entirely new framework is thus urgently needed to bridge the significant gap between classical predictability theory and contemporary forecasting practice.

To bridge this gap, we introduce a novel diagnostic framework grounded in spectral coherence that is computationally efficient, directly aligned with the MSE objective, and provides rich, multi-scale insights. Our framework consists of two core components: 1) **Spectral Coherence Predictability (SCP)**, a per-instance predictability score $\mathcal{P}$ derived from a linear MSE lower bound, $\text{MSE}_{\text{lb}}$. It is computed in $O(N \log N)$ time and is intrinsically consistent with the MSE objective. 2) **Linear Utilization Ratio (LUR)**, a frequency-resolved diagnostic that quantifies how effectively a model exploits linearly predictable information across different spectral bands, enabling fine-grained assessments of under-use, saturation, and beyond-linear gains. With this framework, we aim to shift the time-series evaluation paradigm from simple "model ranking" toward a more profound analysis of "model-data diagnostics," thereby providing clear, actionable guidance for developing more powerful and reliable forecasting models. Through extensive experiments on synthetic and real-world benchmarks, we validate our proposed framework. We demonstrate that our predictability score is well-calibrated and strongly correlates with the empirical performance of state-of-the-art models. Furthermore, our diagnostics uncover the highly time-varying nature of predictability, enabling a fairer, stratified evaluation that uncovers the differential strengths of different architectures, moving beyond the limitations of aggregate scores.

In summary, our contributions are as follows:

- We are the first to systematically address the evaluation ambiguity in modern time-series forecasting by introducing a framework to decouple model error from a time series' intrinsic, irreducible unpredictability.

- We propose a computationally efficient and task-aligned Spectral Coherence Predictability (SCP). It provides a per-instance predictability score with a corresponding error lower bound and a frequency-resolved diagnostic to analyze a model's information utilization.

- Extensive experiments validates this framework's alignment with state-of-the-art models. We then use it to reveal novel insights like "predictability drift" and introduce a stratified evaluation that uncovers the complementary strengths of different models.

## 2 RELATED WORK

Our work is positioned at the intersection of two key research areas: the quantification of sequence predictability and the use of spectral methods for time-series analysis.

**Predictability of Time Series.** Entropy-based notions have long been used to proxy sequence predictability, from Shannon's entropy and entropy rate to variants usable on continuous data (approximate, sample, fuzzy, and permutation entropy) (Shannon, 1948; Pincus, 1991; Richman & Moorman, 2000; Bandt & Pompe, 2002; Garland et al., 2014). Compression-driven estimators (e.g., Lempel–Ziv) provide nonparametric estimates of entropy rate for symbolic, stationary sources (Ziv & Lempel, 1977). These approaches have also been popular in human mobility, where spatio-temporal regularity supports predictability limits under coarse symbolizations (González et al., 2008; Song et al., 2010; Wang et al., 2021). However, they face three key limitations for general forecasting: (i) computational burden the need for discretization of continuous data; (ii) theoretical misalignment with multi-step squared-loss objectives; and (iii) sensitivity of differential entropy to reparameterization and divergence issues in non-stationary settings (Mohammed et al., 2024). Consequently, existing predictability proxies fail to provide a task-aligned and reliable estimate of the achievable error lower bound for multi-step time-series forecasting.

**Spectral Analysis in Time Series Forecasting.** Spectral analysis is a cornerstone of time-series modeling, inspiring many recent deep learning architectures. For instance, Autoformer was designed with an auto-correlation mechanism to discover period-based dependencies efficiently (Wu et al., 2021). FEDformer directly integrates Fourier/Wavelet transforms into attention for frequency-domain computation with reduced complexity (Zhou et al., 2022). TimesNet captures complex multi-periodicity by transforming the 1D time series into a 2D representation for analysis (Wu et al., 2023). These methods all leverage spectral properties to build better models. In contrast, our work uses spectral coherence to build a novel diagnostic framework for analyzing data predictability and evaluating the utilization of existing models.

## 3 PRELIMINARIES

**Problem setup and notation.** We focus on a setting in which an observed sequence is decomposed into a past (history) portion used as input and a future portion serving as ground-truth for evaluation. Formally, a sample consists of a history $\boldsymbol{x} \in \mathbb{R}^N$ and a future $\boldsymbol{y} \in \mathbb{R}^N$ drawn from a distribution $\mathbb{D}$. The goal is to learn a measurable predictor $f : \mathbb{R}^N \to \mathbb{R}^N$ that produces a forecast $\widehat{\boldsymbol{y}} = f(\boldsymbol{x})$. We evaluate predictions with the mean squared error (MSE) per forecast step:

$$\mathrm{MSE}(f; \boldsymbol{x}, \boldsymbol{y}) \;=\; \frac{1}{N} \, \|f(\boldsymbol{x}) - \boldsymbol{y}\|_2^2, \tag{1}$$

where $\|\cdot\|_2$ denotes the Euclidean norm. The learning objective is to minimize the expected risk $\mathbb{E}_{(\boldsymbol{x},\boldsymbol{y})\sim\mathcal{D}}[\mathrm{MSE}(f; \boldsymbol{x}, \boldsymbol{y})]$.

**Intrinsic predictability via Bayes risk.** Under the MSE metric, the risk-minimizing predictor is the conditional expectation $f^\star(\boldsymbol{x}) = \mathbb{E}[\boldsymbol{y} \mid \boldsymbol{x}]$ (Chen et al., 2016). The corresponding minimum achievable risk (Bayes risk) is

$$\mathrm{MSE}^\star \;=\; \mathbb{E}\left[ \frac{1}{N} \, \big\| \boldsymbol{y} - \mathbb{E}[\boldsymbol{y} \mid \boldsymbol{x}] \big\|_2^2 \right]. \tag{2}$$

Using the unconditional variance $\mathrm{Var}(\boldsymbol{y})$ as a baseline, we define intrinsic predictability as the normalized reduction of uncertainty:

$$\mathcal{P}^\star_{xy} \;=\; 1 - \frac{\mathrm{MSE}^\star}{\mathrm{Var}(\boldsymbol{y})}, \tag{3}$$

which coincides with the theoretical coefficient of determination ($R^2$). By the law of total variance, $\mathrm{Var}(\boldsymbol{y}) = \mathrm{MSE}^\star + \mathrm{Var}\big(\mathbb{E}[\boldsymbol{y} \mid \boldsymbol{x}]\big)$, hence $\mathcal{P}^\star \in [0, 1]$. At the extremes, $\mathcal{P}^\star = 1$ if and only if $\mathrm{Var}(\boldsymbol{y} \mid \boldsymbol{x}) = 0$ almost surely, i.e., $\boldsymbol{y}$ is a deterministic function of $\boldsymbol{x}$. Conversely, $\mathcal{P}^\star = 0$ if and only if $\mathrm{Var}\big(\mathbb{E}[\boldsymbol{y} \mid \boldsymbol{x}]\big) = 0$, i.e., the conditional mean $\mathbb{E}[\boldsymbol{y} \mid \boldsymbol{x}]$ is constant and $\boldsymbol{x}$ conveys no information for predicting the mean of $\boldsymbol{y}$.

## 4 METHOD

### 4.1 SPECTRAL COHERENCE PREDICTABILITY

Building on the intrinsic predictability in Eq. (3), we seek a computable surrogate from a single realization by leveraging frequency–domain structure. The resulting Spectral Coherence Predictability (SCP) quantifies how much of the future segment $\boldsymbol{y}$ is linearly explainable by the history segment $\boldsymbol{x}$ across frequencies, and then aggregates the explained and unexplained power.

We operate in the frequency domain using Welch's method. Let $\widehat{S}_{yy}(f)$ and $\widehat{S}_{xx}(f)$ denote the power spectral densities (PSD) of $\boldsymbol{y}$ and $\boldsymbol{x}$, and let $\widehat{S}_{xy}(f)$ denote their cross–power spectral density (CPSD). All spectra are computed on the same discrete Fourier transform (DFT) grid with identical Welch parameters after mean removal. The squared coherence between $\boldsymbol{y}$ and $\boldsymbol{x}$ is

$$\gamma^2_{xy}(f) \;=\; \frac{\big|\widehat{S}_{xy}(f)\big|^2}{\big(\widehat{S}_{xx}(f) + \varepsilon\big)\big(\widehat{S}_{yy}(f) + \varepsilon\big)} \in [0, 1], \tag{4}$$

---

**Algorithm 1** Spectral Coherence Predictability (SCP)

---

**Require:** History $\boldsymbol{x} \in \mathbb{R}^N$, future $\boldsymbol{y} \in \mathbb{R}^N$; Welch parameters (window, segment length, overlap); stability constant $\varepsilon > 0$; optional frequency band $\mathcal{F}_b$.

**Ensure:** MSE linear lower bound $\mathrm{MSE_{lb}}$ and predictability $\mathcal{P}_{xy}$.

1: **Mean removal:** $m_x \leftarrow \mathrm{mean}(x)$, $m_y \leftarrow \mathrm{mean}(y)$; $\Delta^2 \leftarrow (m_y - m_x)^2$; $x \leftarrow x - m_x$, $y \leftarrow y - m_y$.

2: **Welch spectra:** Compute the PSD $\widehat{S}_{xx}(f)$, $\widehat{S}_{yy}(f)$ and the CPSD $\widehat{S}_{xy}(f)$ on the discrete frequency domain $\mathcal{F}$.

3: **Squared coherence:**

$$\gamma^2(f) \leftarrow \frac{|\widehat{S}_{xy}(f)|^2}{\left(\widehat{S}_{xx}(f) + \varepsilon\right)\left(\widehat{S}_{yy}(f) + \varepsilon\right)} \in [0, 1], \quad \forall f \in \mathcal{F}.$$

4: **Residual spectrum:** $\widehat{S}_e(f) \leftarrow \widehat{S}_{yy}(f)\left(1 - \gamma^2(f)\right), \quad \forall f \in \mathcal{F}.$

5: **Frequency set:** $\mathcal{F}_\star \leftarrow \mathcal{F}_b$ if a band $\mathcal{F}_b$ is provided; otherwise $\mathcal{F}_\star \leftarrow \mathcal{F}.$

6: **Aggregate:**

$$\widehat{\mathrm{Var}}(y) \leftarrow \sum_{f \in \mathcal{F}_\star} \widehat{S}_{yy}(f), \qquad \mathrm{MSE_{lb}} \leftarrow \Delta^2 + \sum_{f \in \mathcal{F}_\star} \widehat{S}_e(f).$$

7: **Predictability:** $\mathcal{P}_{xy} \leftarrow 1 - \mathrm{MSE_{lb}}/\widehat{\mathrm{Var}}(y).$

8: **return** $\mathrm{MSE_{lb}}$, $\mathcal{P}_{xy}$ .

---

where $\varepsilon > 0$ is a small term for numerical stability (Mandel & Wolf, 1976; Wang et al., 2019). Interpreting $\gamma_{xy}^2(f)$ as a linearly explained–power ratio, the unexplained (residual) spectrum is

$$\widehat{S}_e(f) = \widehat{S}_{yy}(f)\left(1 - \gamma_{xy}^2(f)\right). \tag{5}$$

Let $\mathcal{F}$ denote the discrete frequency domain under our normalization, so that the total spectral power equals the sample variance, i.e., $\sum_{f \in \mathcal{F}} \widehat{S}_{yy}(f) = \widehat{\mathrm{Var}}(\boldsymbol{y})$. To correct a boundary mean mismatch, we optionally add a boundary mean–shift term $\Delta^2 = \left(\mathrm{mean}(\boldsymbol{y}) - \mathrm{mean}(\boldsymbol{x})\right)^2$. This yields a lower bound on the mean squared error of any linear time–invariant predictor that uses $\boldsymbol{x}$:

$$\mathrm{MSE_{lb}} = \Delta^2 + \sum_{f \in \mathcal{F}} \widehat{S}_e(f), \qquad \widehat{\mathrm{Var}}(\boldsymbol{y}) = \sum_{f \in \mathcal{F}} \widehat{S}_{yy}(f). \tag{6}$$

The SCP estimate of predictability is then

$$\mathcal{P}_{xy} = 1 - \frac{\mathrm{MSE_{lb}}}{\widehat{\mathrm{Var}}(\boldsymbol{y})} \in [0, 1]. \tag{7}$$

Algorithm 1 summarizes the steps. Computationally, with fast Fourier transform and Welch estimation, SCP costs $\mathcal{O}(N \log N)$ per sample. This is substantially lower than matching–based Lempel–Ziv–style predictability estimators, which typically entail at least quadratic–to–cubic time in sequence length (e.g., $\mathcal{O}(N^3)$ in naive implementations) and usually target single–step predictability, whereas SCP yields a multi–step estimate aligned with the evaluation horizon.

**Theoretical interpretation.** If $(\boldsymbol{x}, \boldsymbol{y})$ is jointly Gaussian and wide–sense stationary around the boundary, the Bayes predictor is linear (Ko & Fox, 2009). In this case, Eq. (7) is a consistent estimator of the intrinsic predictability $\mathcal{P}_{xy}^\star$ as the effective sample size grows. For general (possibly non-linear) processes, $\widehat{\mathrm{MSE}}_{\mathrm{lb}}$ lower–bounds the error of any linear time–invariant forecaster using $\boldsymbol{x}$, hence $\mathcal{P}_{xy}$ is a conservative estimate that isolates the explanatory power of stable linear dynamics across frequencies and sets a meaningful baseline for non-linear forecasters.

## 4.2 SLinear Utilization Ratio

Instead of relying on standard metrics like MSE or MAE, which only tell us if a model is accurate, our approach analyzes why it is accurate by comparing its performance to the data's theoretical predictability limit in the frequency domain.

---

**Algorithm 2** Linear Utilization Ratio (LUR)

---

**Require:** History $x \in \mathbb{R}^N$, future $y \in \mathbb{R}^N$, model prediction $\hat{y} \in \mathbb{R}^N$; Welch parameters (window, segment length, overlap); stability $\varepsilon > 0$; optional frequency band $\mathcal{F}_b$.
**Ensure:** Model–explained power $P_{\text{model}}$; linear–explainable power $P_{\text{linear}}$; evaluation ratio LUR.
  1: **Mean removal:** $x \leftarrow x - \text{mean}(x)$; $y \leftarrow y - \text{mean}(y)$; $\hat{y} \leftarrow \hat{y} - \text{mean}(\hat{y})$.
  2: **Welch spectra (same parameters for all):**

$$\widehat{S}_{xx}(f), \ \widehat{S}_{yy}(f), \ \widehat{S}_{\hat{y}\hat{y}}(f), \ \widehat{S}_{xy}(f), \ \widehat{S}_{y\hat{y}}(f), \quad \forall f \in \mathcal{F}.$$

  3: **Coherences:**

$$\gamma_{yx}^2(f) \leftarrow \frac{|\widehat{S}_{yx}(f)|^2}{(\widehat{S}_{yy}(f) + \varepsilon)(\widehat{S}_{xx}(f) + \varepsilon)}, \quad \gamma_{y\hat{y}}^2(f) \leftarrow \frac{|\widehat{S}_{y\hat{y}}(f)|^2}{(\widehat{S}_{yy}(f) + \varepsilon)(\widehat{S}_{\hat{y}\hat{y}}(f) + \varepsilon)}.$$

$\qquad\qquad\qquad\qquad\qquad\qquad\qquad\qquad\qquad\qquad\qquad\qquad\qquad\qquad\qquad$ ▷ Eq. (4) and Eq. (8).
  4: **Frequency set:** $\mathcal{F}_\star \leftarrow \mathcal{F}_b$ if a band $\mathcal{F}_b$ is provided; otherwise $\mathcal{F}_\star \leftarrow \mathcal{F}$.
  5: **Power–weighted aggregation:**

$$P_{\text{model}} \leftarrow \sum_{f \in \mathcal{F}_\star} \gamma_{y\hat{y}}^2(f) \widehat{S}_{yy}(f), \qquad P_{\text{linear}} \leftarrow \sum_{f \in \mathcal{F}_\star} \gamma_{yx}^2(f) \widehat{S}_{yy}(f).$$

  6: **LUR ratio:** $\text{LUR} \leftarrow P_{\text{model}}/P_{\text{linear}}$.
  7: **return** $P_{\text{model}}, \ P_{\text{linear}}, \ \text{LUR}$.

---

Our method, detailed in Algorithm 2, is built on two key quantities: The first key quantity is the linear limit, given by the squared coherence between the future $y$ and the history $x$, $\gamma_{yx}^2(f)$ in Eq. (4), which upper-bounds, at frequency $f$, the fraction of $y$'s power that any linear predictor can explain using $x$. The second quantity is the prediction–target coherence, measuring how much of $y$'s power is actually captured by the model prediction $\hat{y}$ at frequency $f$:

$$\gamma_{y\hat{y}}^2(f) \ = \ \frac{\left|\widehat{S}_{y\hat{y}}(f)\right|^2}{(\widehat{S}_{yy}(f) + \varepsilon)(\widehat{S}_{\hat{y}\hat{y}}(f) + \varepsilon)} \ \in \ [0, 1]. \tag{8}$$

Comparing these two coherences yields a per-frequency diagnosis:

- If $\gamma_{y\hat{y}}^2(f) < \gamma_{yx}^2(f)$, the model under-utilizes linearly predictable information in the history $x$.

- If $\gamma_{y\hat{y}}^2(f) \approx \gamma_{yx}^2(f)$, the model saturates the linear limit of the history $x$.

- If $\gamma_{y\hat{y}}^2(f) > \gamma_{yx}^2(f)$, the model surpasses the linear limit of the history $x$, indicating exploitation of nonlinear patterns or additional, correlated data sources.

To summarize over frequency while prioritizing strong signal regions, we weight coherence by the target power spectrum $\widehat{S}_{yy}(f)$ on the discrete frequency domain $\mathcal{F}$. Define the aggregated powers

$$P_{\text{model}} = \sum_{f \in \mathcal{F}} \gamma_{y\hat{y}}^2(f) \widehat{S}_{yy}(f), \qquad P_{\text{linear}} = \sum_{f \in \mathcal{F}} \gamma_{yx}^2(f) \widehat{S}_{yy}(f), \tag{9}$$

The fraction of target energy that is linearly predictable is

$$\eta_{\text{linear}} \ = \ \frac{P_{\text{linear}}}{\sum_{f \in \mathcal{F}} \widehat{S}_{yy}(f)} \ = \ \frac{P_{\text{linear}}}{\widehat{\text{Var}}(y)}. \tag{10}$$

To further assess how much of this predictable energy the model actually captures, we use the Linear Utilization Ratio (LUR)

$$\text{LUR} \ = \ \frac{P_{\text{model}}}{P_{\text{linear}}} \ \geq 0. \tag{11}$$

When $\text{LUR} < 1$, the model under-exploits linear information available in the history $x$; when $\text{LUR} \approx 1$, it saturates the linear limit; when $\text{LUR} > 1$, it achieves gains beyond linear structure (e.g., nonlinear dynamics or cross-instance inductive patterns learned from the training set).

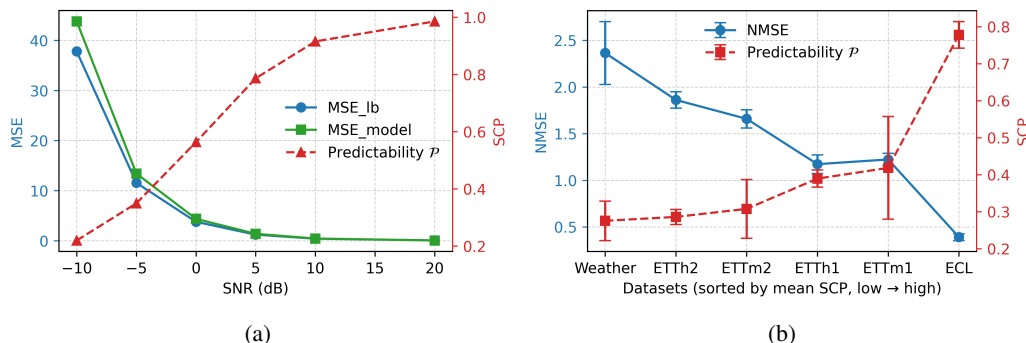

(a)             (b)

Figure 1: Relationship between model error (MSE) and predictability $\mathcal{P}$. (a) MSE of the best linear predictor on a synthetic Gaussian process with varying SNR. (b) Average performance of state-of-the-art prediction models on real datasets. We report normalized MSE (NMSE), obtained by dividing MSE by the corresponding variance.

To analyze behavior across scales, we additionally partition the discrete frequency domain into disjoint bands $\{\mathcal{F}_b\}_{b=1}^B$ (e.g., low/mid/high), using the band partition as in Algorithms 1 and 2. This yields band-limited counterparts $\mathrm{MSE}_{\mathrm{lb},b}$, $\mathcal{P}_{xy,b}$, and $\mathrm{LUR}_b$, which enable localized diagnosis of under-use, saturation, or beyond-linear gains within each frequency band.

## 5 EXPERIMENTS

This section empirically evaluates our metric and diagnostics across synthetic and real-world settings, guided by three questions: **Q1:** Does our proposed metric accurately reflect the intrinsic predictability of time series data (Secs. 5.1 and 5.2)? **Q2:** What novel insights and observations can be derived from this new metric (Sec. 5.3)? **Q3:** How can our metric be leveraged to enable more comprehensive evaluation methodologies for forecasting models (Secs. 5.4 and 5.5)?

### 5.1 TOY STUDY

We first validate our proposed predictability score $\mathcal{P}$ and theoretical error lower bound $\mathrm{MSE}_{\mathrm{lb}}$ in a controlled, synthetic environment. Our setup consists of a Gaussian process with additive noise at varying SNRs, on which we evaluate an optimal linear forecaster (Fig. 1a). For each SNR, we report the model's test MSE, the estimated linear lower bound $\mathrm{MSE}_{\mathrm{lb}}$, and the corresponding predictability $\mathcal{P}$. As noise decreases (higher SNR), $\mathcal{P}$ increases monotonically toward one, and the model MSE approaches $\mathrm{MSE}_{\mathrm{lb}}$. Across all SNRs, $\mathrm{MSE}_{\mathrm{lb}}$ remains strictly below the realized MSE, confirming it is a valid lower bound; the shrinking gap at high SNR indicates that the linear model saturates the data-implied limit when the process is nearly noise-free. It validates both the calibration (monotonic response to noise) and the tightness (small bound–error gap in the linear regime) of our estimates.

### 5.2 ALIGNING PREDICTABILITY AND FORECASTING PERFORMANCE

To ensure a fair and rigorous comparison, we evaluate all baselines under a strictly controlled setup: (i) the forecast horizon and the history length are fixed and identical across models; (ii) the common "drop–last" heuristic is disabled. For correlation analyses, we report the Pearson coefficient $R$ between each model's empirical prediction MSE and the theoretically estimated $\mathrm{MSE}_{\mathrm{lb}}$, averaged over all variables and samples in the test set to provide a holistic summary.

As shown in Table 1, $\mathrm{MSE}_{\mathrm{lb}}$ strongly correlates with the empirical MSE of SOTA models across datasets, with Pearson correlations typically $R \geq 0.8$. This alignment indicates that the bound accurately predicts where forecasting is intrinsically easier or harder, in agreement with realized model accuracy. To visualize this relationship, Fig. 2 plots iTransformer's MSE against $\mathrm{MSE}_{\mathrm{lb}}$ for each channel in ECL and Weather datasets, revealing a near-linear trend. Collectively, these dataset- and channel-level results validate the effectiveness of our $\mathrm{MSE}_{\mathrm{lb}}$ estimator. They also highlight substantial within-dataset heterogeneity: predictability and MSE can vary markedly across channels.

Table 1: Long-term multivariate forecasting results. We report MSE, MAE, NMSE for forecasting lengths equal to history length $N \in \{96, 192, 336, 720\}$ under an identical protocol (same preprocessing and no drop-last). **Bold** marks the best (lowest) MSE/MAE per column across models. *Average* rows give the column-wise mean across models. Predictability reports the per-task linear MSE lower bound ($\mathrm{MSE}_{lb}$) and SCP $\mathcal{P}$ (higher is easier).

| Models | Metric | ETTh1 | | | | ETTh2 | | | | ETTm1 | | | | ETTm2 | | | | ECL | | | | Weather | | | |
|---|---|---|---|---|---|---|---|---|---|---|---|---|---|---|---|---|---|---|---|---|---|---|---|---|---|
| | | 96 | 192 | 336 | 720 | 96 | 192 | 336 | 720 | 96 | 192 | 336 | 720 | 96 | 192 | 336 | 720 | 96 | 192 | 336 | 720 | 96 | 192 | 336 | 720 |
| iTransformer (Liu et al., 2024) | MSE | 0.387 | 0.441 | 0.471 | 0.700 | 0.301 | 0.381 | 0.426 | 0.425 | 0.342 | 0.345 | 0.379 | 0.448 | 0.186 | 0.254 | 0.289 | 0.382 | **0.148** | 0.156 | 0.167 | **0.194** | 0.176 | 0.214 | 0.255 | 0.353 |
| | MAE | 0.405 | 0.440 | 0.464 | 0.608 | 0.350 | 0.405 | 0.438 | 0.455 | 0.377 | 0.378 | 0.403 | 0.449 | 0.272 | 0.319 | 0.341 | 0.407 | **0.239** | 0.250 | 0.266 | **0.287** | 0.216 | 0.255 | 0.290 | 0.357 |
| | NMSE | 1.121 | 1.095 | 1.056 | 1.292 | 1.598 | 1.829 | 1.563 | 1.432 | 1.340 | 1.167 | 1.140 | 1.159 | 1.609 | 1.696 | 1.595 | 1.917 | 1.288 | 1.268 | 1.280 | 1.319 | 2.629 | 2.353 | 2.032 | 2.150 |
| | R | 0.844 | 0.876 | 0.899 | 0.747 | 0.907 | 0.883 | 0.877 | 0.842 | 0.845 | 0.782 | 0.803 | 0.869 | 0.868 | 0.834 | 0.898 | 0.840 | 0.723 | 0.778 | 0.821 | 0.826 | 0.900 | 0.876 | 0.801 | 0.824 |
| TimeMixer (Wang et al., 2024) | MSE | **0.381** | 0.440 | 0.482 | 0.631 | **0.289** | 0.377 | 0.390 | 0.435 | **0.322** | 0.337 | 0.380 | 0.484 | **0.176** | 0.231 | 0.280 | 0.376 | 0.153 | 0.155 | 0.172 | 0.214 | **0.169** | **0.198** | 0.249 | 0.347 |
| | MAE | 0.400 | 0.434 | 0.460 | 0.561 | **0.340** | 0.406 | 0.423 | 0.458 | 0.359 | 0.372 | 0.396 | 0.469 | 0.259 | 0.296 | 0.332 | 0.390 | 0.245 | 0.244 | 0.264 | 0.310 | **0.215** | **0.242** | 0.291 | 0.355 |
| | NMSE | 1.131 | 1.069 | 0.848 | 1.155 | 0.916 | 1.724 | 1.524 | 1.446 | 1.303 | 1.105 | 1.155 | 1.202 | 1.493 | 1.485 | 1.498 | 1.689 | 0.282 | 0.274 | 0.286 | 0.334 | 2.602 | 2.161 | 2.107 | 2.162 |
| | R | 0.815 | 0.889 | 0.848 | 0.793 | 0.916 | 0.801 | 0.909 | 0.906 | 0.829 | 0.781 | 0.752 | 0.798 | 0.843 | 0.867 | 0.910 | 0.732 | 0.706 | 0.607 | 0.682 | 0.887 | 0.911 | 0.862 | 0.885 | 0.852 |
| DLinear (Zeng et al., 2023) | MSE | 0.383 | **0.422** | 0.447 | 0.507 | 0.329 | 0.375 | 0.463 | 0.740 | 0.346 | 0.342 | 0.372 | **0.415** | 0.187 | 0.242 | 0.278 | 0.374 | 0.195 | 0.163 | 0.169 | 0.197 | 0.197 | 0.225 | 0.263 | 0.315 |
| | MAE | **0.396** | **0.421** | 0.448 | 0.517 | 0.380 | 0.410 | 0.472 | 0.609 | 0.374 | 0.369 | **0.389** | **0.415** | 0.281 | 0.315 | 0.338 | 0.406 | 0.277 | 0.259 | 0.268 | 0.295 | 0.255 | 0.282 | 0.314 | 0.354 |
| | NMSE | 1.214 | 1.143 | 1.310 | 1.782 | 2.927 | 2.067 | 2.728 | 3.896 | 1.327 | 1.205 | 1.208 | 1.121 | 1.541 | 1.580 | 1.318 | 1.380 | 0.868 | 0.678 | 0.574 | 0.684 | 3.507 | 2.899 | 2.474 | 2.069 |
| | R | 0.869 | 0.878 | 0.872 | 0.804 | 0.845 | 0.880 | 0.798 | 0.439 | 0.868 | 0.887 | 0.819 | 0.884 | 0.833 | 0.813 | 0.910 | 0.902 | 0.880 | 0.867 | 0.909 | 0.864 | 0.924 | 0.931 | 0.911 | 0.923 |
| PatchTST (Nie et al., 2023) | MSE | 0.391 | 0.429 | **0.436** | **0.465** | 0.293 | **0.357** | **0.363** | **0.406** | 0.322 | 0.328 | **0.365** | 0.417 | 0.177 | **0.230** | **0.276** | **0.356** | 0.167 | **0.151** | 0.167 | 0.212 | 0.176 | 0.202 | **0.247** | **0.309** |
| | MAE | 0.403 | 0.426 | **0.440** | **0.482** | 0.342 | **0.387** | **0.402** | **0.442** | 0.358 | **0.364** | 0.390 | 0.419 | **0.258** | **0.294** | **0.329** | **0.385** | 0.252 | **0.242** | 0.258 | 0.304 | 0.217 | 0.243 | **0.281** | **0.331** |
| | NMSE | 1.074 | 1.065 | 0.982 | 1.037 | 1.541 | 1.580 | 1.318 | 1.380 | 1.245 | 1.093 | 1.130 | 1.079 | 1.549 | 1.471 | 1.485 | 1.612 | 0.320 | 0.267 | 0.292 | 0.339 | 2.777 | 2.159 | 1.987 | 1.818 |
| | R | 0.849 | 0.900 | 0.916 | 0.900 | 0.918 | 0.901 | 0.907 | 0.866 | 0.867 | 0.803 | 0.789 | 0.861 | 0.834 | 0.845 | 0.892 | 0.917 | 0.777 | 0.761 | 0.739 | 0.900 | 0.931 | 0.873 | 0.850 | 0.900 |
| TimesNet (Wu et al., 2023) | MSE | 0.389 | 0.460 | 0.487 | 0.641 | 0.337 | 0.405 | 0.399 | 0.447 | 0.334 | 0.414 | 0.429 | 0.482 | 0.189 | 0.239 | 0.320 | 0.383 | 0.168 | 0.189 | 0.209 | 0.305 | 0.169 | 0.220 | 0.272 | 0.334 |
| | MAE | 0.412 | 0.456 | 0.477 | 0.582 | 0.371 | 0.424 | 0.433 | 0.463 | 0.375 | 0.414 | 0.434 | 0.477 | 0.266 | 0.306 | 0.357 | 0.408 | 0.272 | 0.291 | 0.308 | 0.382 | 0.219 | 0.265 | 0.301 | 0.350 |
| | NMSE | 1.205 | 1.227 | 1.113 | 1.341 | 1.952 | 2.115 | 1.556 | 1.527 | 1.391 | 1.424 | 1.352 | 1.361 | 1.714 | 1.621 | 1.933 | 2.012 | 0.316 | 0.345 | 0.350 | 0.482 | 2.555 | 2.531 | 2.294 | 2.035 |
| | R | 0.869 | 0.881 | 0.915 | 0.876 | 0.886 | 0.920 | 0.911 | 0.864 | 0.813 | 0.652 | 0.741 | 0.835 | 0.904 | 0.908 | 0.863 | 0.909 | 0.735 | 0.809 | 0.737 | 0.969 | 0.912 | 0.903 | 0.868 | 0.865 |
| Average | MSE | 0.386 | 0.438 | 0.465 | 0.589 | 0.310 | 0.379 | 0.408 | 0.491 | 0.333 | 0.353 | 0.385 | 0.449 | 0.183 | 0.239 | 0.289 | 0.374 | 0.166 | 0.163 | 0.177 | 0.224 | 0.177 | 0.212 | 0.257 | 0.332 |
| | MAE | 0.403 | 0.435 | 0.458 | 0.550 | 0.357 | 0.406 | 0.434 | 0.485 | 0.369 | 0.379 | 0.402 | 0.446 | 0.267 | 0.306 | 0.339 | 0.399 | 0.257 | 0.257 | 0.273 | 0.316 | 0.224 | 0.257 | 0.295 | 0.349 |
| | NMSE | 1.149 | 1.120 | 1.105 | 1.321 | 1.912 | 1.863 | 1.738 | 1.936 | 1.321 | 1.199 | 1.193 | 1.184 | 1.608 | 1.599 | 1.626 | 1.805 | 0.415 | 0.366 | 0.356 | 0.432 | 2.814 | 2.421 | 2.179 | 2.047 |
| **Predictability** | $\mathrm{MSE}_{lb}$ | 0.354 | 0.417 | 0.404 | 0.412 | 0.298 | 0.360 | 0.309 | 0.356 | 0.228 | 0.307 | 0.513 | 0.436 | 0.175 | 0.248 | 0.295 | 0.361 | 0.239 | 0.219 | 0.167 | 0.241 | 0.185 | 0.244 | 0.278 | 0.317 |
| | $\mathcal{P}$ | 0.422 | 0.379 | 0.368 | 0.389 | 0.305 | 0.270 | 0.302 | 0.267 | 0.590 | 0.460 | 0.268 | 0.356 | 0.415 | 0.315 | 0.230 | 0.271 | 0.751 | 0.755 | 0.829 | 0.777 | 0.345 | 0.240 | 0.228 | 0.289 |

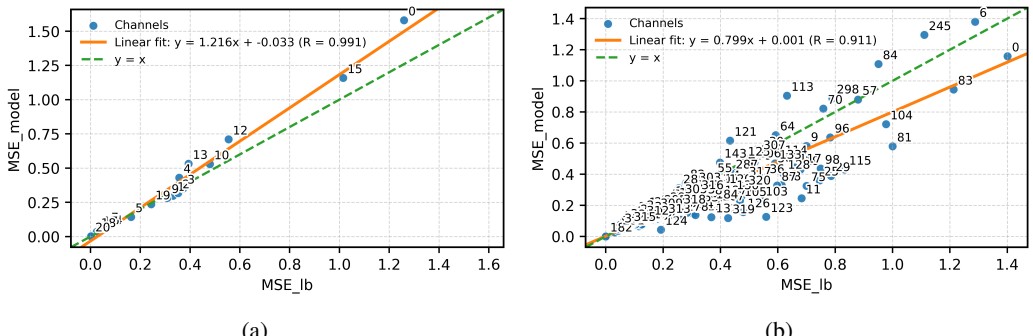

| (a) | (b) |
|---|---|

Figure 2: Per-variable scatter plots on Weather (a) and ECL (b) comparing the estimated MSE lower bound ($\mathrm{MSE}_{lb}$) with iTransformer's prediction error $(\mathrm{MSE})_{model}$.

To compare predictability across datasets, we aggregate over horizons and report, for each dataset, the mean±std of SCP $\mathcal{P}$ and realized NMSE (Fig. 1b). The relationship is strongly inverse: datasets with higher $\mathcal{P}$ tend to exhibit lower NMSE on average. In particular, ECL shows the highest $\mathcal{P}$ and the lowest NMSE (easiest to forecast), whereas Weather exhibits the lowest $\mathcal{P}$ and the highest NMSE (hardest). This ordering indicates that $\mathcal{P}$ ranks dataset difficulty in a manner that aligns with realized errors.

Overall, no single architecture dominates across all datasets and horizons—an expected outcome in predictability-limited settings that, as shown later, arises from time- and band-dependent variability in the exploitable history $x$ that challenges both linear and nonlinear representations.

## 5.3 TIME-VARYING PREDICTABILITY

Standard evaluation metrics, which average performance over an entire test set, implicitly assume that the forecasting task is statistically stationary. However, this assumption often fails for time series tasks. Focusing on a single channel, we move sample by sample, plotting two key metrics in parallel: the model's instantaneous error (MSE) and our measure of the data's predictable energy, decomposed by frequency.

The results, shown in Fig. 3, are striking. We find that a model's performance is not random but is tightly coupled to the instantaneous predictable energy in the data. This reveals that predictability is not a fixed property of a dataset; it varies sharply from one moment to the next. This phenomenon,

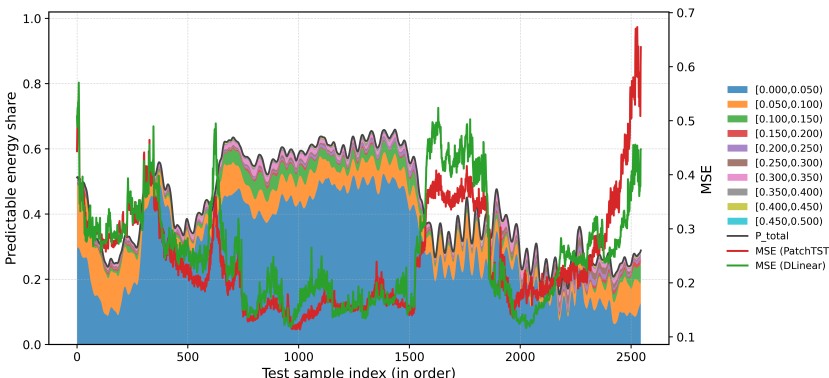

Figure 3: ETTh1 test set, channel 1, horizon $N = 336$. Relationship between per-sample linearly predictable energy (decomposed by frequency band as a share of total) and the corresponding MSE of DLinear and PatchTST.

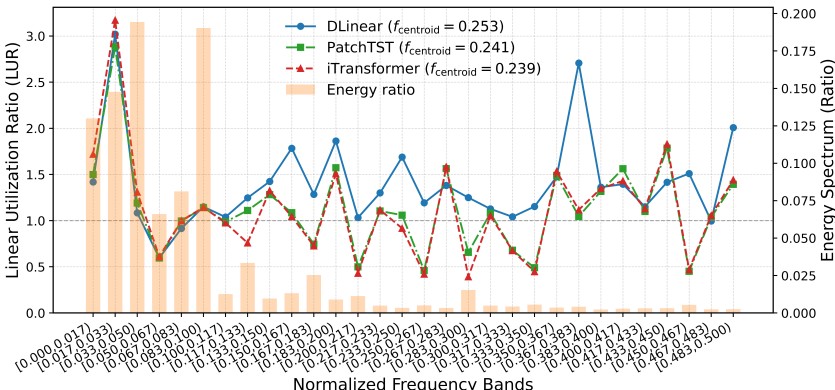

Figure 4: Band-wise analysis on ETTh1, representative channel. Normalized energy shares and LUR across frequency bands for three models.

which we term predictability drift, means the forecasting task should be viewed not as a single problem, but as a time-varying mixture of easy and hard regimes.

When the total predictable share is low, or when the dominant predictable bands shift rapidly, MSE spikes. These observations explain why aggregate test-set statistics often fail to discriminate models. Fluctuations in sample-level predictability always dominate realized error. They also motivate finer-grained evaluation protocols, such as band-wise diagnostics and predictability-aware reporting, which enable grouping evaluations conditioned on data characteristics and provide actionable guidance for model development.

### 5.4 BAND-WISE EVALUATION

To gain a more granular understanding, we use the Linear Utilization Ratio (LUR) to analyze model behavior in the frequency domain. We decompose the test signal into distinct frequency bands and, for each band, measure both the proportion of the signal's total energy and the LUR for several representative models.

Figure 4 reveals the distinct strategies employed by different model architectures. In the five low-frequency bands, which contain the majority of the signal's energy, all three models exhibit a similar pattern: their LURs closely track the energy distribution. This indicates that all models correctly identify and attempt to capture these dominant, high-energy components of the signal. Within these critical bands, however, the PatchTST and iTransformer achieve a slightly higher LUR than DLinear, suggesting they are more efficient at extracting information from these primary signal components.

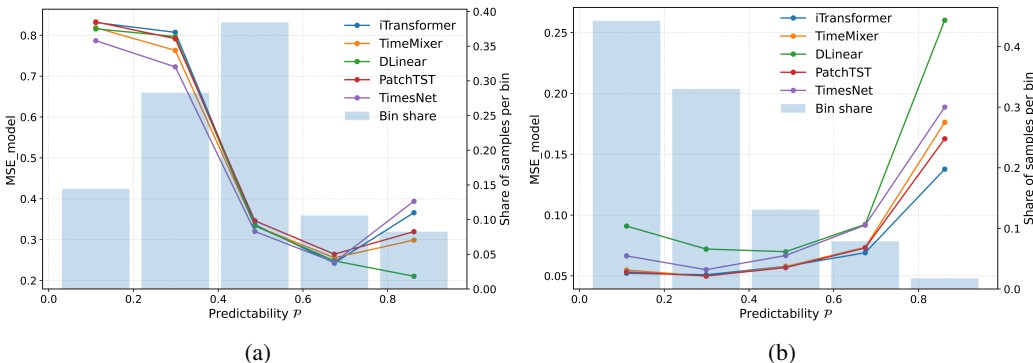

Figure 5: ETTh1 dataset with forecasting length $N = 96$: per-channel evaluation stratified by predictability $\mathcal{P}$. Samples are grouped into equal-width $\mathcal{P}$ bins; each point reports the mean MSE within the bin.

A more significant divergence appears in the higher-frequency bands. Here, DLinear maintains a significantly higher LUR than both PatchTST and iTransformer. This finding exposes the fundamental difference in their underlying mechanisms. DLinear, being a simple linear model, acts like a broadband filter, attempting to capture information indiscriminately across the entire spectrum. In contrast, Transformer-based models like PatchTST and iTransformer function as frequency-selective filters. They intelligently focus their capacity on the high-energy, low-frequency bands where the most predictable signal resides, while largely ignoring the less informative, high-frequency noise. This demonstrates their more sophisticated inductive bias for typical time-series data.

### 5.5 PREDICTABILITY-AWARE EVALUATION

Although most models achieve a similar MSE of $\approx 0.38$ on the ETTh1 dataset (horizon $N = 96$), this average score obscures critical performance differences. By stratifying the test set by our predictability score $\mathcal{P}$, we uncover these hidden behaviors (Fig. 5). In Fig. 5a, nonlinear forecasters (e.g., TimesNet) perform better in the low-$\mathcal{P}$ bin (hard samples), whereas DLinear outperforms in the high-$\mathcal{P}$ (easy samples) bin. In Fig. 5b, where samples concentrate in the low-$\mathcal{P}$ region, iTransformer, TimeMixer, and PatchTST exhibit convergent performance, while DLinear is capacity-limited on hard cases and tends to underfit easy cases due to their low ratio. These contrasts highlight complementary strengths across architectures: nonlinear models excel when the linear predictable signal is scarce, whereas simple linear models are highly competitive when predictability is high.

## 6 CONCLUSION AND FUTURE WORK

This work confronts a fundamental flaw in time series forecasting evaluation: the conflation of model error with intrinsic data difficulty. We introduced a predictability-aligned diagnostic framework, centered on the Spectral Coherence Predictability (SCP) score and the Linear Utilization Ratio (LUR), to disentangle these factors. Our analysis not only provided the first systematic evidence of "predictability drift" but also revealed that different architectures possess complementary strengths, a nuance lost in aggregate metrics. Ultimately, our contribution is a call to shift the community's focus from simplistic model rankings to a more insightful, diagnostic approach. This paradigm change fosters a deeper understanding of model behavior, enables fairer comparisons, and paves the way for the development of more robust and targeted forecasting solutions. Looking ahead, our framework opens the door to new data-aware learning paradigms. For instance, the SCP score can guide curriculum learning strategies by ordering data by difficulty, while the LUR could serve as a novel regularizer, paving the way for more adaptive and robust forecasting systems.

ETHICS STATEMENT

This work does not introduce new data collection involving human subjects. All experiments use publicly available datasets (ETT, Weather, ECL) under their original licenses and terms of use; no sensitive attributes (e.g., protected classes or personally identifiable information) are inferred or used. Our analyses operate on aggregated forecasting errors and frequency-domain summaries, not on individual-level predictions tied to identity or behavior.

REPRODUCIBILITY STATEMENT

We aim for full reproducibility and have released all code, configurations, and scripts to reproduce our results, including data preparation, backbone training/evaluation, and the SCP/LUR diagnostics. All code and data can be found in `https://anonymous.4open.science/r/TS_Predictability-C8B7`.

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

## A  USAGE OF LLMS

We utilized a large language model (LLM) to proofread and improve the grammatical clarity of this manuscript. All scientific ideas, methodologies, and conclusions presented are the original work of the authors.

## B  EXPERIMENTAL SETUP

### B.1  TOY STUDY

We synthesize a multiband Gaussian process and evaluate linear forecasting under controlled band-limited noise. Signals are split into history/future with boundary-paired segments of length $N_p = 512$ from a total length $2N$ with $N = 1024$. Power spectra and coherences are estimated via Welch's method (Hann window) with $n_{\text{perseg}} = 256$ and $n_{\text{overlap}} = 128$. The forecaster is a causal FIR least-squares filter (Wiener approximation) of length $L_{\text{FIR}} = 64$ with ridge $10^{-6}$. The base process has four spectral peaks at rFFT bins $\{32, 96, 192, 384\}$ with widths $\{6, 10, 14, 18\}$ and amplitudes $\{3.0, 2.0, 1.5, 1.0\}$. We sweep noise levels $\{0, 0.25, 0.5, 1.0, 2.0, 4.0\}$ on a single band (index 1 in the plot) and average over 3 trials, reporting model MSE, $\text{MSE}_{\text{lb}}$, and SCP.

### B.2  BACKBONE

We evaluate seven state-of-the-art backbones spanning diverse architectures: Transformer-based (iTransformer (Liu et al., 2024), PatchTST (Nie et al., 2023)), MLP-based (DLinear (Zeng et al., 2023), TimeMixer (Wang et al., 2024)), and CNN-based (TimesNet (Wu et al., 2023)). We adopt the official implementations and recommended hyperparameters from their repositories. To ensure strict comparability, we fix the forecasting horizon and enforce equal input and output lengths for all backbones (no "drop-last"), using identical preprocessing and dataset splits across models.

### B.3  DATASETS

We conduct experiments on eight standard long-horizon multivariate forecasting benchmarks: ETTh1, ETTh2, ETTm1, ETTm2, ECL, Weather, Traffic, and ILI. These datasets cover electricity systems, meteorology, transportation, and epidemiology, and are widely used in recent long-horizon time series forecasting studies.

- **ETT (Electricity Transformer Temperature).** The ETT benchmarks record seven variables from electricity transformers collected in two regions between July 2016 and July 2018. Variants differ in sampling granularity: "h" denotes hourly data and "m" denotes 15-minute data; suffixes "1/2" indicate the two regions (ETTh1/ETTh2, ETTm1/ETTm2) (Zhou et al., 2021). Following standard practice, we treat each variant as a separate multivariate dataset.

- **ECL (Electricity Consuming Load).** ECL provides hourly electricity consumption for 321 clients from 2012 to 2014 (Li et al., 2019). It is a large-scale dataset with strong daily and weekly periodicities and substantial cross-series correlations.

- **Weather.** Weather contains 21 meteorological variables (for example, temperature, humidity, wind speed) measured every 10 minutes in 2020 at the Max Planck Biogeochemistry Institute's weather station (Zhou et al., 2021). It provides a medium-scale, high-frequency benchmark with strong diurnal and seasonal patterns.

- **Traffic.** Traffic consists of hourly road occupancy rates collected from 862 loop sensors on San Francisco Bay Area freeways over two years (Wu et al., 2021). The series are bounded in $[0, 1]$ and exhibit strong rush-hour and weekday/weekend patterns, making the dataset a high-dimensional transportation benchmark.

- **ILI (Influenza-Like Illness).** ILI contains weekly statistics on influenza-like illness in the United States, including the number of patients and the ILI ratio aggregated over multiple regions (Lai et al., 2018). Compared with the other datasets, ILI is much shorter and lower frequency, with pronounced annual seasonality and relatively high noise, and is therefore evaluated with shorter prediction horizons.

---

**Algorithm 3** Multivariate Spectral Coherence Predictability ($\text{SCP}_{\text{multi}}$)

---

**Require:** History $\boldsymbol{x} \in \mathbb{R}^{d_x \times N}$, future $\boldsymbol{y} \in \mathbb{R}^{d_y \times N}$; Welch parameters (window, length, overlap); stability constant $\varepsilon > 0$; optional frequency band $\mathcal{F}_b$.
**Ensure:** Multivariate MSE lower bound $\text{MSE}_{\text{lb}}^{\text{multi}}$ and predictability $\mathcal{P}_{xy}^{\text{multi}}$.

 1: **Mean removal:** $\Delta^2 \leftarrow \|\boldsymbol{\mu}_y - \boldsymbol{\mu}_x\|_2^2$; $\boldsymbol{x} \leftarrow \boldsymbol{x} - \boldsymbol{\mu}_x$, $\boldsymbol{y} \leftarrow \boldsymbol{y} - \boldsymbol{\mu}_y$.
 2: **Welch spectra:** Compute matrix-valued PSDs $\widehat{S}_{xx}(f)$, $\widehat{S}_{yy}(f)$ and CPSD $\widehat{S}_{xy}(f)$ on $\mathcal{F}$; set $\widehat{S}_{yx}(f) = \widehat{S}_{xy}(f)^H$.
 3: **Multichannel Wiener spectra:**

$$\widehat{S}_{\hat{y}\hat{y}}(f) = \widehat{S}_{yx}(f)\big(\widehat{S}_{xx}(f) + \varepsilon I_{d_x}\big)^{-1}\widehat{S}_{xy}(f), \quad \widehat{S}_e(f) = \widehat{S}_{yy}(f) - \widehat{S}_{\hat{y}\hat{y}}(f).$$

 4: **Frequency set:** $\mathcal{F}_\star \leftarrow \mathcal{F}_b$ if $\mathcal{F}_b$ is provided; otherwise $\mathcal{F}_\star \leftarrow \mathcal{F}$.
 5: **Aggregate:**

$$\widehat{\text{Var}}(\boldsymbol{y}) \leftarrow \sum_{f \in \mathcal{F}_\star} \text{tr}\,\widehat{S}_{yy}(f), \qquad \text{MSE}_{\text{lb}}^{\text{multi}} \leftarrow \Delta^2 + \sum_{f \in \mathcal{F}_\star} \text{tr}\,\widehat{S}_e(f).$$

 6: **Predictability:** $\mathcal{P}_{xy}^{\text{multi}} \leftarrow 1 - \text{MSE}_{\text{lb}}^{\text{multi}}/\widehat{\text{Var}}(\boldsymbol{y})$.
 7: **return** $\text{MSE}_{\text{lb}}^{\text{multi}}$, $\mathcal{P}_{xy}^{\text{multi}}$.

---

### B.4 TIME-TO-FREQUENCY

For each test instance we split the sequence into history $\boldsymbol{x}$ and future $\boldsymbol{y}$ (equal lengths by default), remove sample means, and estimate power and cross-spectra with Welch's method using identical settings for $\boldsymbol{x}, \boldsymbol{y}$, and (when available) $\hat{\boldsymbol{y}}$: Hann window with length $n_{\text{win}} = \lfloor 0.25N \rfloor$, 50% overlap, and real FFT on the one-sided grid $\mathcal{F}$ with variance-preserving normalization. We form squared coherences with a small ridge $\varepsilon$ for stability, compute the residual spectrum to obtain the linear lower bound $\text{MSE}_{\text{lb}}$ and predictability $\mathcal{P} = 1 - \text{MSE}_{\text{lb}}/\text{Var}(\boldsymbol{y})$, and derive utilization metrics (global or band-wise) via target-power–weighted aggregation of $\gamma_{y\hat{y}}^2$ and $\gamma_{yx}^2$.

## C METHOD EXTENSIONS

### C.1 MULTIVARIATE EXTENSION

#### C.1.1 MULTIVARIATE SCP

We extend the univariate SCP in Sec. 4.1 to multivariate histories and futures with input dimensionality $d_x$ and output dimensionality $d_y$. Let $\boldsymbol{x}_t \in \mathbb{R}^{d_x}$ and $\boldsymbol{y}_t \in \mathbb{R}^{d_y}$ denote a length-$N$ history–future pair, and let $\boldsymbol{x} = (\boldsymbol{x}_1, \ldots, \boldsymbol{x}_N) \in \mathbb{R}^{d_x \times N}$, $\boldsymbol{y} = (\boldsymbol{y}_1, \ldots, \boldsymbol{y}_N) \in \mathbb{R}^{d_y \times N}$. Using Welch's method with shared parameters for all components, we compute multivariate power spectral density (PSD) and cross–power spectral density (CPSD) matrices on a discrete frequency grid $\mathcal{F}$:

$$\widehat{S}_{xx}(f) \in \mathbb{C}^{d_x \times d_x}, \qquad \widehat{S}_{yy}(f) \in \mathbb{C}^{d_y \times d_y}, \qquad \widehat{S}_{xy}(f) \in \mathbb{C}^{d_x \times d_y}, \qquad (12)$$

and set $\widehat{S}_{yx}(f) = \widehat{S}_{xy}(f)^H$, where $H$ denotes the Hermitian transpose.

At frequency $f$, the optimal linear time–invariant predictor from $\boldsymbol{x}$ to $\boldsymbol{y}$ in the least-squares sense has transfer matrix

$$H(f) = \widehat{S}_{yx}(f)\big(\widehat{S}_{xx}(f) + \varepsilon I_{d_x}\big)^{-1}, \qquad (13)$$

where $\varepsilon > 0$ is the same Tikhonov regularization as in Eq. (4), and $I_{d_x}$ is the $d_x \times d_x$ identity matrix. The spectrum of the linearly predictable component of $\boldsymbol{y}$ is then

$$\widehat{S}_{\hat{y}\hat{y}}(f) = H(f)\,\widehat{S}_{xx}(f)\,H(f)^H = \widehat{S}_{yx}(f)\big(\widehat{S}_{xx}(f) + \varepsilon I_{d_x}\big)^{-1}\widehat{S}_{xy}(f) \in \mathbb{C}^{d_y \times d_y}. \qquad (14)$$

In the scalar case $d_x = d_y = 1$, Eq. (14) reduces to $\widehat{S}_{\hat{y}\hat{y}}(f) = |\widehat{S}_{xy}(f)|^2/(\widehat{S}_{xx}(f) + \varepsilon)$, which coincides with the univariate expression $\gamma_{xy}^2(f)\,\widehat{S}_{yy}(f)$ in Eq. (4).

---

**Algorithm 4** Multivariate Linear Utilization Ratio (LUR$_{\text{multi}}$)

---

**Require:** History $\boldsymbol{x} \in \mathbb{R}^{d_x \times N}$, future $\boldsymbol{y} \in \mathbb{R}^{d_y \times N}$, prediction $\hat{\boldsymbol{y}} \in \mathbb{R}^{d_y \times N}$; Welch parameters (window, length, overlap); stability $\varepsilon > 0$; optional band $\mathcal{F}_b$.

**Ensure:** Multivariate model–explained power $P_{\text{model}}$, linear–explainable power $P_{\text{linear}}$, and utilization ratio $\text{LUR}^{\text{multi}}$.

1: **Mean removal:** $\boldsymbol{x} \leftarrow \boldsymbol{x} - \text{mean}(\boldsymbol{x})$; $\boldsymbol{y} \leftarrow \boldsymbol{y} - \text{mean}(\boldsymbol{y})$; $\hat{\boldsymbol{y}} \leftarrow \hat{\boldsymbol{y}} - \text{mean}(\hat{\boldsymbol{y}})$.

2: **Welch spectra:** Compute $\widehat{S}_{xx}(f)$, $\widehat{S}_{yy}(f)$, $\widehat{S}_{\hat{y}\hat{y}}^{\text{pred}}(f)$, $\widehat{S}_{xy}(f)$, $\widehat{S}_{y\hat{y}}(f)$ on $\mathcal{F}$; set $\widehat{S}_{yx}(f) = \widehat{S}_{xy}(f)^H$ and $\widehat{S}_{\hat{y}y}(f) = \widehat{S}_{y\hat{y}}(f)^H$.

3: **Linear limit (per frequency):**
$$\widehat{S}_{\hat{y}\hat{y}}(f) \leftarrow \widehat{S}_{yx}(f)\big(\widehat{S}_{xx}(f) + \varepsilon I_{d_x}\big)^{-1}\widehat{S}_{xy}(f), \quad P_{\text{linear}}(f) \leftarrow \text{tr}\,\widehat{S}_{\hat{y}\hat{y}}(f).$$

4: **Model–explained power (per frequency):**
$$P_{\text{model}}(f) \leftarrow \text{tr}\Big(\widehat{S}_{y\hat{y}}(f)\big(\widehat{S}_{\hat{y}\hat{y}}^{\text{pred}}(f) + \varepsilon I_{d_y}\big)^{-1}\widehat{S}_{\hat{y}y}(f)\Big).$$

5: **Frequency set:** $\mathcal{F}_\star \leftarrow \mathcal{F}_b$ if a band $\mathcal{F}_b$ is provided; otherwise $\mathcal{F}_\star \leftarrow \mathcal{F}$.

6: **Aggregation:**
$$P_{\text{linear}} \leftarrow \sum_{f \in \mathcal{F}_\star} P_{\text{linear}}(f), \qquad P_{\text{model}} \leftarrow \sum_{f \in \mathcal{F}_\star} P_{\text{model}}(f).$$

7: **LUR ratio:** $\text{LUR}^{\text{multi}} \leftarrow P_{\text{model}}/P_{\text{linear}}$.

8: **return** $P_{\text{model}}$, $P_{\text{linear}}$, $\text{LUR}^{\text{multi}}$.

---

The residual spectrum matrix is
$$\widehat{S}_e(f) = \widehat{S}_{yy}(f) - \widehat{S}_{\hat{y}\hat{y}}(f), \qquad \forall f \in \mathcal{F}. \tag{15}$$

Since $\widehat{S}_{\hat{y}\hat{y}}(f)$ is the least-squares projection of $\widehat{S}_{yy}(f)$ onto the subspace linearly spanned by $\boldsymbol{x}$, the true residual spectrum is positive semidefinite, and the regularization $\varepsilon I_{d_x}$ stabilizes this property numerically. Let the estimated total variance (total power) of $\boldsymbol{y}$ be the trace–aggregated spectrum
$$\widehat{\text{Var}}(\boldsymbol{y}) = \sum_{f \in \mathcal{F}} \text{tr}\,\widehat{S}_{yy}(f), \tag{16}$$

where $\text{tr}(\cdot)$ denotes the matrix trace. Using the same frequency grid, the multivariate MSE lower bound induced by linear time–invariant predictors is
$$\text{MSE}_{\text{lb}}^{\text{multi}} = \Delta^2 + \sum_{f \in \mathcal{F}} \text{tr}\,\widehat{S}_e(f), \tag{17}$$

where $\Delta^2$ is the same boundary mean–shift term as in the univariate case, generalized to the $(d_x, d_y)$-dimensional setting.

The multivariate SCP is defined by normalizing the residual energy as in Eq. (7):
$$\mathcal{P}_{xy}^{\text{multi}} = 1 - \frac{\text{MSE}_{\text{lb}}^{\text{multi}}}{\widehat{\text{Var}}(\boldsymbol{y})} \in [0, 1]. \tag{18}$$

When $d_x = d_y = 1$, Eq. (18) reduces exactly to the univariate SCP in Eq. (7).

### C.1.2 MULTIVARIATE LUR

The spectrum of the linearly predictable component in Eq. (14) induces the linear–explainable power
$$P_{\text{linear}}(f) = \text{tr}\,\widehat{S}_{\hat{y}\hat{y}}(f) = \text{tr}\Big(\widehat{S}_{yx}(f)\big(\widehat{S}_{xx}(f) + \varepsilon I_{d_x}\big)^{-1}\widehat{S}_{xy}(f)\Big). \tag{19}$$

For the model, we form the auto- and cross-spectra of the prediction,
$$\widehat{S}_{\hat{y}\hat{y}}^{\text{pred}}(f) \in \mathbb{C}^{d_y \times d_y}, \qquad \widehat{S}_{y\hat{y}}(f) \in \mathbb{C}^{d_y \times d_y}, \qquad \widehat{S}_{\hat{y}y}(f) = \widehat{S}_{y\hat{y}}(f)^H, \tag{20}$$

and define the model–explained power via the optimal linear projection of $\boldsymbol{y}$ onto the subspace spanned by $\hat{\boldsymbol{y}}$:

$$P_{\text{model}}(f) = \text{tr}\left(\widehat{S}_{y\hat{y}}(f)\big(\widehat{S}_{\hat{y}\hat{y}}^{\text{pred}}(f) + \varepsilon I_{d_y}\big)^{-1}\widehat{S}_{\hat{y}y}(f)\right). \tag{21}$$

Aggregating over the discrete frequency domain $\mathcal{F}$,

$$P_{\text{linear}} = \sum_{f \in \mathcal{F}} P_{\text{linear}}(f), \qquad P_{\text{model}} = \sum_{f \in \mathcal{F}} P_{\text{model}}(f), \tag{22}$$

and normalizing as in Sec. 4.2 gives the multivariate linear utilization ratio

$$\text{LUR}^{\text{multi}} = \frac{P_{\text{model}}}{P_{\text{linear}}}. \tag{23}$$

When $d_x = d_y = 1$, these expressions reduce to the univariate definitions of $P_{\text{linear}}$, $P_{\text{model}}$, and LUR.

## C.2 NONLINEAR EXTENSION

The SCP framework is linear by construction: it characterizes the best linear time–invariant (LTI) predictor in the original observation space. To relax this restriction while preserving the same spectral machinery, we introduce a nonlinear feature map

$$\phi : \mathbb{R}^{d_x} \to \mathbb{R}^{d_z}, \qquad \boldsymbol{z}_t = \phi(\boldsymbol{x}_t) \in \mathbb{R}^{d_z}, \tag{24}$$

and apply multivariate SCP in the resulting feature space. We then form the feature sequence $\boldsymbol{z} = (\boldsymbol{z}_1, \ldots, \boldsymbol{z}_N) \in \mathbb{R}^{d_z \times N}$. The map $\phi$ can use explicit nonlinear features (e.g., polynomial expansions or a shallow encoder), or be defined implicitly by a kernel $k(\boldsymbol{x}, \boldsymbol{x}') = \langle \phi(\boldsymbol{x}), \phi(\boldsymbol{x}') \rangle$ in an RKHS. Using the same Welch configuration as before, we estimate the multivariate spectra

$$\widehat{S}_{zz}(f) \in \mathbb{C}^{d_z \times d_z}, \qquad \widehat{S}_{yy}(f) \in \mathbb{C}^{d_y \times d_y}, \qquad \widehat{S}_{yz}(f) \in \mathbb{C}^{d_y \times d_z}, \tag{25}$$

and set $\widehat{S}_{zy}(f) = \widehat{S}_{yz}(f)^H$.

In feature space, the optimal LTI predictor of $\boldsymbol{y}$ from $\boldsymbol{z}$ takes the same form as the multivariate Wiener filter in Eq. (13), but with $(\boldsymbol{x}, \widehat{S}_{xx})$ replaced by $(\boldsymbol{z}, \widehat{S}_{zz})$:

$$H_\phi(f) = \widehat{S}_{yz}(f)\left(\widehat{S}_{zz}(f) + \varepsilon I_{d_z}\right)^{-1}, \tag{26}$$

where $\varepsilon > 0$ is the same Tikhonov regularization as before and $I_{d_z}$ is the $d_z \times d_z$ identity. The spectrum of the component of $\boldsymbol{y}$ that is linearly predictable from the nonlinear features is

$$\widehat{S}_{\hat{y}\hat{y}}^{\text{ker}}(f) = H_\phi(f)\,\widehat{S}_{zz}(f)\,H_\phi(f)^H = \widehat{S}_{yz}(f)\left(\widehat{S}_{zz}(f) + \varepsilon I_{d_z}\right)^{-1}\widehat{S}_{zy}(f) \in \mathbb{C}^{d_y \times d_y}. \tag{27}$$

When $\phi$ is the identity map ($d_z = d_x$ and $\boldsymbol{z}_t = \boldsymbol{x}_t$), Eq. (27) reduces to the multivariate linear spectrum in Eq. (14).

The residual spectrum under the feature-space predictor is

$$\widehat{S}_e^{\text{ker}}(f) = \widehat{S}_{yy}(f) - \widehat{S}_{\hat{y}\hat{y}}^{\text{ker}}(f), \qquad \forall f \in \mathcal{F}, \tag{28}$$

which is positive semidefinite in the ideal (population) setting. Aggregating as in Eq. (16), the total variance of $\boldsymbol{y}$ and the corresponding nonlinear MSE lower bound are

$$\widehat{\text{Var}}(\boldsymbol{y}) = \sum_{f \in \mathcal{F}} \text{tr}\,\widehat{S}_{yy}(f), \qquad \text{MSE}_{\text{lb}}^{\text{ker}} = \Delta^2 + \sum_{f \in \mathcal{F}} \text{tr}\,\widehat{S}_e^{\text{ker}}(f), \tag{29}$$

where $\Delta^2$ is the same boundary mean–shift term used in Eq. (17), applied to the multivariate setting.

The nonlinear SCP is then obtained by normalizing the feature-space residual:

$$\mathcal{P}_{xy}^{\text{nonlin}} = 1 - \frac{\text{MSE}_{\text{lb}}^{\text{ker}}}{\widehat{\text{Var}}(\boldsymbol{y})}. \tag{30}$$

This quantity measures the fraction of future variance that is explainable by LTI predictors acting on the chosen nonlinear feature representation, providing a feature-dependent notion of nonlinear predictability.

### C.3 BEYOND EVALUATION

The predictability scores from SCP (and its multivariate / nonlinear variants) $\mathcal{P}_{xy} \in [0, 1]$ can be used not only for post-hoc analysis, but also to shape how data are selected and organized during training.

**Hard-example mining** For a dataset $\{(\boldsymbol{x}^{(i)}, \boldsymbol{y}^{(i)})\}_{i=1}^M$ with per-sample predictability $\mathcal{P}^{(i)} \equiv \mathcal{P}_{xy}(\boldsymbol{x}^{(i)}, \boldsymbol{y}^{(i)})$, we can directly use $\mathcal{P}^{(i)}$ to reweight the loss:

$$L = \sum_{i=1}^M w^{(i)} \ell(\hat{\boldsymbol{y}}^{(i)}, \boldsymbol{y}^{(i)}), \qquad w^{(i)} \propto (\mathcal{P}^{(i)})^\alpha, \tag{31}$$

with $\alpha < 0$. This up-weights intrinsically predictable segments and down-weights near-unpredictable ones that mainly contain irreducible noise.

**Curriculum learning** The same scores induce a simple curriculum over data difficulty. Let $\{\tau_s\}_{s=1}^S$ be a decreasing sequence of thresholds, $\tau_1 > \tau_2 > \cdots > \tau_S$. At stage $s$, we restrict training to

$$\mathcal{D}_s = \{ i : \mathcal{P}^{(i)} \geq \tau_s \}, \tag{32}$$

i.e., the model first sees highly predictable segments, and progressively incorporates samples with lower $\mathcal{P}^{(i)}$ as $s$ increases.

**Anomaly detection and change points** On a time series stream, we compute predictability over a sliding window ending at time $t$, for example $\mathcal{P}_t = \mathcal{P}_{xy}(\boldsymbol{x}_{t-N+1:t}, \boldsymbol{y}_{t+1:t+N})$. Let $\mu_{\mathcal{P}}, \sigma_{\mathcal{P}}$ be the mean and standard deviation of $\mathcal{P}_t$ on a reference (normal) period. We flag $t$ as anomalous when

$$|\mathcal{P}_t - \mu_{\mathcal{P}}| > \kappa \sigma_{\mathcal{P}}, \tag{33}$$

with a chosen threshold $\kappa > 0$. Sudden drops or spikes in $\mathcal{P}_t$ indicate changes in intrinsic predictability, and thus potential regime shifts or anomalous behavior.

### C.4 COMPARISON WITH TIME-DOMAIN CORRELATION DIAGNOSTICS

Classical time-domain tools such as the autocorrelation function (ACF) provide a convenient way to visualize second-order structure by plotting correlation as a function of lag. ACF is particularly useful for qualitatively assessing periodicity and dependence decay. However, it is primarily a descriptive tool for the self-correlation of a single series. In particular, ACF does not directly quantify how well a future window can be linearly predicted from a past window under the MSE objective, especially in the multi-horizon, multivariate setting we consider.

In contrast, our SCP/LUR framework is explicitly constructed around the past–future prediction task. SCP is derived from the cross-spectral density and coherence between the history and future segments, and measures the fraction of the future variance that is linearly explainable from the observed history, yielding an MSE-aligned notion of intrinsic predictability. LUR further decomposes this explainable energy across frequency bands, revealing which parts of the spectrum are well captured or systematically missed by a given model.

A simple example illustrates the difference between naive time-domain correlation and spectral coherence. Consider two noiseless signals

$$x_t = \sin(\omega_0 t), \qquad y_t = \cos(\omega_0 t).$$

Here $y_t$ is a phase-shifted version of $x_t$, obtained by a linear time-invariant transformation. In other words, $y$ is perfectly linearly predictable from $x$.

If we look only at the zero-lag Pearson correlation

$$\rho_{xy}(0) = \text{corr}(x_t, y_t),$$

and average over many periods by treating $\theta = \omega_0 t$ as uniform on $[0, 2\pi]$, we obtain

$$\mathbb{E}[\sin\theta \cos\theta] = 0,$$

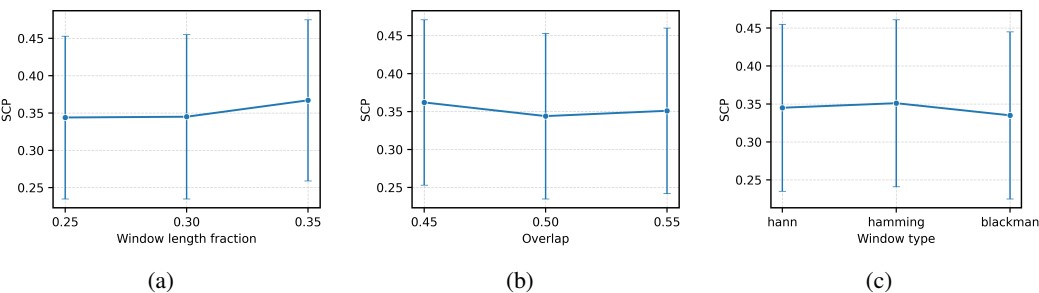

Figure 6: Sensitivity of SCP to Welch hyperparameters on the Weather dataset. Error bars indicate the mean and standard deviation computed over 10000 samples.

hence $\rho_{xy}(0) = 0$. A time-domain diagnostic based solely on zero-lag correlation would therefore suggest that $x$ and $y$ are "unrelated", even though $y$ is deterministically generated from $x$ by a linear filter.

In the frequency domain, both $x$ and $y$ have all their energy concentrated at the same frequency $\omega_0$. Their cross-spectrum at $\omega_0$ differs only by a constant phase factor, so the squared coherence

$$\gamma^2(\omega_0) = \frac{|S_{xy}(\omega_0)|^2}{S_{xx}(\omega_0)\, S_{yy}(\omega_0)}$$

evaluates to $\gamma^2(\omega_0) = 1$. In our framework, this implies a linear MSE lower bound of zero and SCP equal to one: the spectral diagnostic correctly recognizes that $y$ is fully predictable from $x$ despite the phase shift. This example highlights that simple time-domain summaries such as zero-lag correlation can miss strong linear predictability when phase shifts or distributed lags are present, whereas coherence (and thus SCP) aggregates information over all lags at each frequency and is invariant to such shifts.

# D SUPPLEMENTARY EXPERIMENTS

## D.1 SENSITIVITY ANALYSIS

This section presents two complementary sensitivity studies. First, we investigate how the SCP metric responds to the hyperparameters of the Welch coherence estimator. Second, we analyze whether conclusions drawn from LUR remain consistent under different frequency-band partitions.

### D.1.1 SENSITIVITY TO WELCH PARAMETERS

We assess the sensitivity of SCP to the Welch coherence hyperparameters. On the Weather dataset, with the forecast horizon fixed to $N = 96$, we vary three factors: the window-length fraction $L_w/N$, the overlap ratio $\rho$, and the tapering window (Hann, Hamming, Blackman). We adopt $(L_w/N, \rho, \text{window}) = (0.25, 0.5, \text{Hann})$ as the default configuration, and vary one hyperparameter at a time while keeping the others fixed.

As shown in Figure 6 and Table 2, the mean SCP changes only slightly across the entire parameter range, indicating that the estimator remains stable under standard spectral settings. The relatively large standard deviations stem from the data rather than the estimator: different temporal regions exhibit distinct intrinsic predictability, consistent with the non-stationary structure illustrated in Figure 3.

### D.1.2 SENSITIVITY TO FREQUENCY-BAND PARTITIONING

As illustrated in Figures 7 and 8, changing the band boundaries affects the absolute LUR values within each band, but the qualitative conclusions remain unchanged. Across all configurations, iTransformer consistently achieves higher LUR in the low-frequency region where most signal energy concentrates, whereas DLinear performs better in the high-frequency bands. The frequency

Table 2: SCP and linear MSE lower bound ($\text{MSE}_{\text{lb}}$) under different Welch configurations.

| Parameter | Value | SCP (mean $\pm$ std) | $\text{MSE}_{\text{lb}}$ (mean $\pm$ std) |
|---|---|---|---|
| Window-length fraction ($L_w/N$) | 0.25 | $0.344 \pm 0.109$ | $0.186 \pm 0.133$ |
| | 0.30 | $0.345 \pm 0.110$ | $0.186 \pm 0.133$ |
| | 0.35 | $0.367 \pm 0.108$ | $0.183 \pm 0.133$ |
| Overlap ($\rho$) | 0.45 | $0.362 \pm 0.109$ | $0.184 \pm 0.134$ |
| | 0.50 | $0.344 \pm 0.109$ | $0.185 \pm 0.133$ |
| | 0.55 | $0.351 \pm 0.109$ | $0.185 \pm 0.135$ |
| Window type | Hann | $0.345 \pm 0.110$ | $0.186 \pm 0.134$ |
| | Hamming | $0.351 \pm 0.110$ | $0.185 \pm 0.133$ |
| | Blackman | $0.335 \pm 0.110$ | $0.187 \pm 0.135$ |

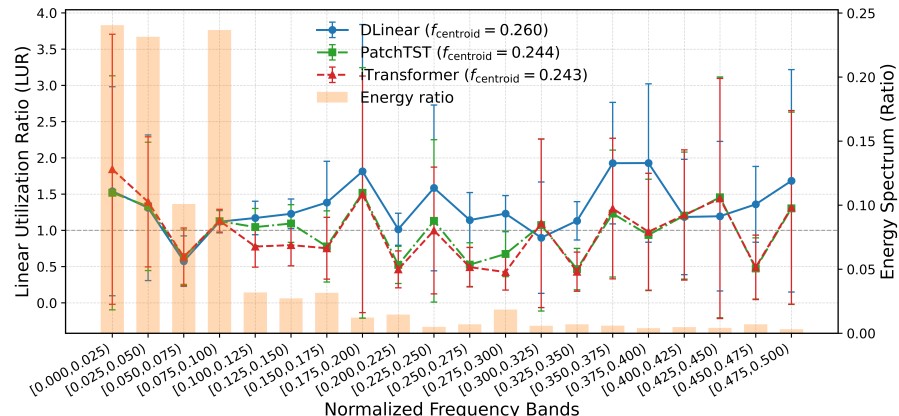

Figure 7: Band-wise normalized energy and LUR on ETTh1 using a 20-band partition.

centroid $f_{\text{centroid}}$ exhibits the same ordering: DLinear attains the largest centroid, while PatchTST and iTransformer remain close.

## D.2 MULTIVARIATE PREDICTABILITY

To validate that our metric captures multivariate predictability, we construct a controlled synthetic example. The input is a $d_x$-dimensional process $\boldsymbol{x}(n) \in \mathbb{R}^{d_x}$ and the target is scalar ($d_y = 1$). We set $d_x = 6$, $d_y = 1$, sequence length $N = 1024$, and number of independent sequences $N_{\text{samples}} = 640$.

For each input dimension $i \in \{1, \ldots, d_x\}$ we generate a sinusoid $x_i(n) = \sin\big(2\pi f_i n + \phi_i\big)$ for $n = 0, \ldots, N-1$, with distinct frequencies $f_i = k_i/L_w$ for a Welch window length $L_w = 128$ and $(k_1, \ldots, k_6) = (3, 5, 7, 9, 11, 13)$, aligned to discrete Fourier bins. The phases are drawn independently as $\phi_i \sim \text{Unif}[0, 2\pi]$ for each $i$ and each sequence. The target signal is defined as a noisy sum of all input components, $y(n) = \sum_{i=1}^{d_x} x_i(n) + \epsilon(n)$, with $\epsilon(n) \sim \mathcal{N}(0, 0.05)$, so that most of the target energy is linearly generated by the $d_x$ inputs.

For each $m \in \{1, \ldots, d_x\}$ we only reveal the first $m$ input dimensions $(x_1, \ldots, x_m)$ and compute the resulting multivariate SCP $\mathcal{P}_{\text{lin}}^{\text{multi}}(m)$ and multivariate MSE lower bound $\text{MSE}_{\text{lb}}^{\text{multi}}(m)$. Both quantities are averaged over the $N_{\text{samples}}$ sequences, and we also report their empirical standard deviations. The numerical results are summarized in Table 3, and the corresponding curves are shown in Fig. 9.

The results exhibit a clear, approximately monotonic trend. As the number of observed input dimensions $m$ increases, the multivariate SCP $\mathcal{P}_{\text{lin}}^{\text{multi}}(m)$ rises almost linearly, while the multivariate $\text{MSE}_{\text{lb}}^{\text{multi}}(m)$ decreases accordingly. As $m$ approaches $d_x$, $\mathcal{P}_{\text{lin}}^{\text{multi}}(m)$ approaches the ideal predictability implied by the signal-to-noise ratio (but does not reach 1 due to spectral estimation error and the injected noise), and $\text{MSE}_{\text{lb}}^{\text{multi}}(m)$ correspondingly approaches zero. Taken together, the

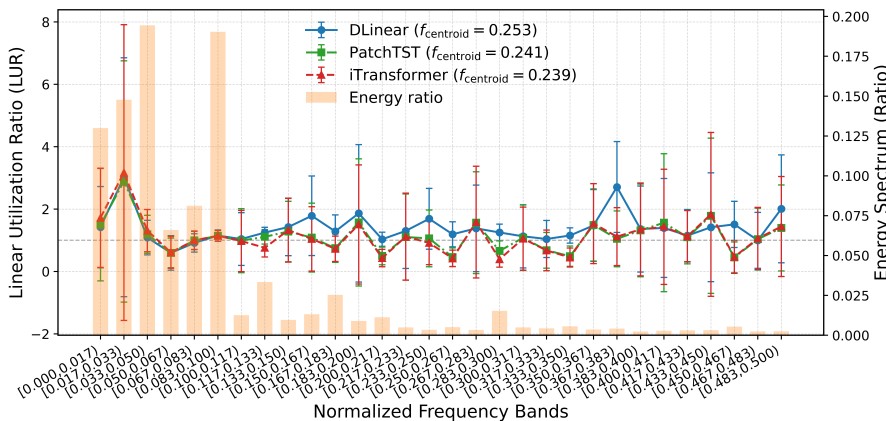

Figure 8: Band-wise normalized energy and LUR on ETTh1 using a 30-band partition.

Table 3: Multivariate SCP and MSE lower bound versus the number of observed input dimensions $m$ in the synthetic sinusoid mixture experiment.

| $m$ | $\mathcal{P}_{\text{lin}}^{\text{multi}}$ (mean $\pm$ std) | $\text{MSE}_{\text{lb}}^{\text{multi}}$ (mean $\pm$ std) |
|---|---|---|
| 1 | $0.117 \pm 0.121$ | $0.486 \pm 0.067$ |
| 2 | $0.208 \pm 0.145$ | $0.436 \pm 0.080$ |
| 3 | $0.360 \pm 0.130$ | $0.353 \pm 0.072$ |
| 4 | $0.523 \pm 0.169$ | $0.263 \pm 0.094$ |
| 5 | $0.662 \pm 0.151$ | $0.186 \pm 0.083$ |
| 6 | $0.848 \pm 0.030$ | $0.084 \pm 0.017$ |

table and figure confirm that multivariate SCP faithfully tracks the gain in predictability contributed by additional informative input dimensions.

### D.3 VARIABLE HISTORY WINDOW ($N_x \neq N_y$)

Let $N_x$ and $N_y$ denote the history and future lengths used for SCP. The construction only requires that a contiguous history–future pair exists around the boundary; $N_x$ and $N_y$ need not coincide. Given a Welch segment length $L_{\text{w}}$ and overlap ratio $\text{overlap} \in [0, 1)$, the effective shift between consecutive segments is

$$\Delta = L_{\text{w}}(1 - \text{overlap}),$$

and the approximate number of Welch segments for a sequence of length $N$ is

$$K(N; L_{\text{w}}, \text{overlap}) \approx \left\lfloor \frac{N - L_{\text{w}}}{L_{\text{w}}(1 - \text{overlap})} \right\rfloor + 1. \tag{34}$$

As a concrete example, consider a history window $N_x = 192$, a longer future horizon $N_y = 336$, and a Welch window $L_{\text{w}} = 64$. With an overlap of $\text{overlap} = 0.5$, the hop size is $\Delta = 64(1-0.5) = 32$, and the corresponding numbers of Welch segments are

$$K_x \approx K(192; 64, 0.5) = \left\lfloor \tfrac{192-64}{32} \right\rfloor + 1 = 5, \qquad K_y \approx K(336; 64, 0.5) = \left\lfloor \tfrac{336-64}{32} \right\rfloor + 1 = 9.$$

For each segment we form windowed signals $\boldsymbol{x}_k(t)$ and $\boldsymbol{y}_k(t)$ of length $L_{\text{w}}$, compute their discrete Fourier transforms $X_k(f)$ and $Y_k(f)$, and define the auto-spectra by Welch averaging

$$\widehat{S}_{xx}(f) = \frac{1}{K_x} \sum_{k=1}^{K_x} |X_k(f)|^2, \qquad \widehat{S}_{yy}(f) = \frac{1}{K_y} \sum_{k=1}^{K_y} |Y_k(f)|^2.$$

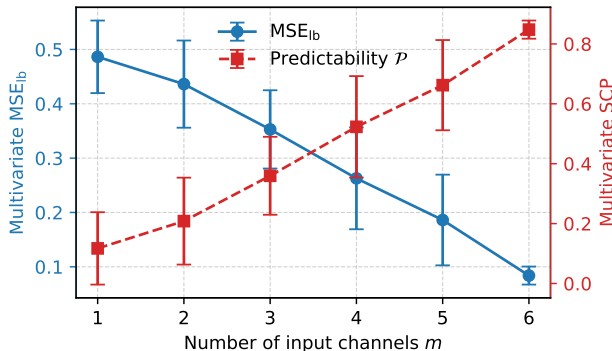

Figure 9: Multivariate SCP and MSE lower bound versus the number of input dimensions $m$ in the synthetic sinusoid mixture experiment. Error bars indicate standard deviation across $N_{\text{samples}}$ independent sequences.

Table 4: Effect of history window length $N_x$ on model error (MSE, $\text{MSE}_{\text{lb}}$) and correlation $R$ on ETTh1 (prediction horizon $N_y = 336$).

|  |  | History length $N_x$ | | | |
| --- | --- | --- | --- | --- | --- |
| Model | Metric | 96 | 192 | 336 | 720 |
| iTransformer | MSE | 0.491 | 0.479 | 0.471 | 0.480 |
|  | $R$ | 0.808 | 0.842 | 0.899 | 0.832 |
| DLinear | MSE | 0.491 | 0.480 | 0.447 | 0.449 |
|  | $R$ | 0.788 | 0.840 | 0.872 | 0.853 |
| Predictibility | $\text{MSE}_{\text{lb}}$ | 0.431 | 0.433 | 0.404 | 0.424 |

The cross-spectrum is computed on the aligned history–future portion at the boundary: we use the last $K_{\text{pair}} = \min(K_x, K_y)$ segments from the history and the first $K_{\text{pair}}$ segments from the future, denote their transforms by $X_k^{(\text{hist})}(f)$ and $Y_k^{(\text{fut})}(f)$, and set

$$\widehat{S}_{xy}(f) = \frac{1}{K_{\text{pair}}} \sum_{k=1}^{K_{\text{pair}}} X_k^{(\text{hist})}(f) \, \overline{Y_k^{(\text{fut})}(f)}.$$

Thus the shorter side effectively limits $K_{\text{pair}}$ and hence the stability of $\widehat{S}_{xy}(f)$, while additional segments on the longer side primarily reduce the variance of the marginal auto-spectra.

We conduct the experiment on ETTh1 with prediction horizon $N_y = 336$ and history lengths $N_x \in \{96, 192, 336, 720\}$, using both iTransformer and DLinear. Table 4 summarizes the results.

On ETTh1, the MSE of both models varies as the history length changes. However, Across all history lengths, the correlation $R$ between SCP and per-sample errors remains high (typically $R \geq 0.80$). This shows that SCP provides a robust measure of intrinsic predictability that is largely insensitive to the exact history length, even though the models' absolute accuracy can be sensitive to this choice.

### D.4 ADDITIONAL DATASET EVALUATION

To complement the main experiments, we further evaluate our framework on the full set of eight long-horizon multivariate forecasting benchmarks: ETTh1, ETTh2, ETTm1, ETTm2, ECL, Weather, Traffic, and ILI. These datasets span electricity, meteorology, transportation, and epidemiology, and cover a wide range of dimensionalities and sampling frequencies. Table 5 summarizes the basic statistics and forecasting horizon settings used in this extended evaluation.

Table 5: Detailed descriptions of the datasets used in our extended evaluation. "Number of variables" gives the dimensionality of each dataset. "Dataset size" denotes the total number of time points in the training, validation, and test splits. "Prediction length" denotes the forecasting horizon; four horizon settings are used for each dataset. "Frequency" is the sampling interval.

| Dataset | Dim | Prediction Length | Dataset Size | Frequency | Information |
|---|---|---|---|---|---|
| ETTh1, ETTh2 | 7 | $\{96, 192, 336, 720\}$ | (8545, 2881, 2881) | Hourly | Electricity |
| ETTm1, ETTm2 | 7 | $\{96, 192, 336, 720\}$ | (34465, 11521, 11521) | 15min | Electricity |
| ECL | 321 | $\{96, 192, 336, 720\}$ | (18317, 2633, 5261) | Hourly | Electricity |
| Weather | 21 | $\{96, 192, 336, 720\}$ | (36792, 5271, 10540) | 10min | Weather |
| Traffic | 862 | $\{96, 192, 336, 720\}$ | (12185, 1757, 3509) | Hourly | Transportation |
| ILI | 7 | $\{60, 72\}$ | (617, 74, 170) | Weekly | Epidemiology |

Table 6: Long-term multivariate forecasting results on Traffic and ILI datasets. We report MSE, MAE, NMSE, and R. **Bold** marks the best (lowest MSE/MAE) per column across models. *Average* rows give the column-wise mean across models. Predictability reports the per-task linear MSE lower bound ($MSE_{lb}$) and SCP $\mathcal{P}$ (higher is easier).

| Models | Metric | Traffic | | | | ILI | |
|---|---|---|---|---|---|---|---|
| | | 96 | 192 | 336 | 720 | 60 | 72 |
| iTransformer (Liu et al., 2024) | MSE | **0.394** | **0.385** | **0.388** | **0.416** | 2.001 | 2.186 |
| | MAE | **0.269** | 0.269 | 0.274 | 0.290 | 0.954 | 1.033 |
| | NMSE | 0.303 | 0.302 | 0.265 | 0.273 | 0.806 | 1.032 |
| | R | 0.849 | 0.917 | 0.959 | 0.968 | 0.687 | 0.847 |
| TimeMixer (Wang et al., 2024) | MSE | 0.485 | 0.423 | 0.407 | 0.437 | 2.272 | **1.928** |
| | MAE | 0.319 | 0.285 | 0.275 | 0.297 | 0.977 | **0.938** |
| | NMSE | 0.376 | 0.327 | 0.275 | 0.292 | 1.055 | 0.714 |
| | R | 0.919 | 0.939 | 0.971 | 0.960 | 0.783 | 0.781 |
| DLinear (Zeng et al., 2023) | MSE | 0.649 | 0.459 | 0.436 | 0.450 | 2.671 | 2.661 |
| | MAE | 0.396 | 0.305 | 0.296 | 0.306 | 1.083 | 1.114 |
| | NMSE | 0.541 | 0.370 | 0.305 | 0.302 | 1.077 | 1.099 |
| | R | 0.899 | 0.929 | 0.970 | 0.968 | 0.850 | 0.919 |
| PatchTST (Nie et al., 2023) | MSE | 0.451 | 0.402 | 0.401 | 0.434 | **1.758** | 2.010 |
| | MAE | 0.288 | **0.263** | **0.267** | **0.289** | **0.863** | 0.948 |
| | NMSE | 0.356 | 0.321 | 0.277 | 0.288 | 0.753 | 0.792 |
| | R | 0.893 | 0.920 | 0.966 | 0.965 | 0.675 | 0.852 |
| TimesNet (Wu et al., 2023) | MSE | 0.606 | 0.608 | 0.630 | 0.672 | 2.160 | 1.994 |
| | MAE | 0.327 | 0.329 | 0.347 | 0.357 | 0.961 | 0.974 |
| | NMSE | 0.370 | 0.376 | 0.331 | 0.335 | 0.866 | 0.697 |
| | R | 0.960 | 0.969 | 0.946 | 0.983 | 0.820 | 0.742 |
| Average | MSE | 0.517 | 0.455 | 0.452 | 0.482 | 2.172 | 2.156 |
| | MAE | 0.320 | 0.290 | 0.292 | 0.308 | 0.968 | 1.001 |
| | NMSE | 0.389 | 0.339 | 0.291 | 0.298 | 0.911 | 0.867 |
| | R | 0.904 | 0.935 | 0.962 | 0.969 | 0.763 | 0.828 |
| **Predictability** | $MSE_{lb}$ | 0.803 | 0.616 | 0.400 | 0.636 | 2.151 | 2.681 |
| | $\mathcal{P}$ | 0.514 | 0.619 | 0.760 | 0.610 | 0.560 | 0.466 |

Table 6 reports detailed long-horizon multivariate forecasting results on the Traffic and Illness datasets for five representative architectures under a matched-information protocol: the history length equals the prediction horizon ($N \in \{96, 192, 336, 720\}$ for Traffic and $N \in \{60, 72\}$ for ILI), with identical preprocessing and no drop-last. We report MSE, MAE, normalized MSE (NMSE), and correlation coefficient $R$, together with the linear MSE lower bound $MSE_{lb}$ and SCP-based predictability $\mathcal{P}$ for each task.

On Traffic, iTransformer consistently achieves the lowest MSE across all horizons. On Illness, TimeMixer and PatchTST achieve better accuracy than the other baselines. Across both datasets, the SCP and linear MSE lower bound remain well aligned with the empirical results, indicating that our predictability-aware metrics continue to agree with, and help interpret, standard error–based evaluations.

# E    LIMITATIONS AND DISCUSSION

Our estimates rely on windowed Welch spectra and an assumption of local second–order stationarity within each window. These conditions are used to derive a tight connection between SCP and the Bayes–optimal MSE lower bound in the linear setting. In practice, however, our real–world benchmarks are visibly nonstationary, with predictability drifting over time (e.g., Fig. 4), yet SCP

remains well aligned with observed forecasting errors, including those of nonlinear architectures. This supports the use of SCP/LUR as robust diagnostics of intrinsic difficulty even when the formal tightness guarantees no longer strictly hold.

Moreover, the framework is not restricted to univariate series: our matrix-based formulation naturally extends SCP/LUR to multivariate settings, and nonlinear preprocessing can be applied to map raw observations into a feature space before computing SCP, so that the linear prediction lower bound is evaluated in that transformed space. A full treatment of conditional and partial coherence structures remains an interesting direction for future work.

Finally, we view predictability-aware learning schemes as a promising next step. Extending SCP to guide active data selection, curriculum-style scheduling across predictability regimes, and hard-sample mining based on $\mathcal{P}$ and LUR are natural applications of the framework beyond the purely diagnostic use studied in this paper.

