# OpenReview forum: "Beyond Model Ranking: Predictability-Aligned Evaluation for Time Series Forecasting"
_ICLR.cc/2026/Conference — Submitted to ICLR 2026_

### Official Review · Reviewer_dD4o · 2025-10-30

**Soundness:** 3
**Presentation:** 3
**Contribution:** 3
**Rating:** 6
**Confidence:** 3

**Summary:**

The paper proposes a forecasting evaluation framework built on two spectral diagnostics: SCP, which estimates how much of the future is linearly predictable from the past, and LUR, which measures how much of that predictable portion a given model actually captures. Experiments illustrate that predictable energy correlates with errors and that certain models better exploit specific bands, offering practical guidance for improving forecasters. The approach is computationally light and broadly applicable across models, making it a useful diagnostic layer atop standard forecasting benchmarks.

**Strengths:**

- Clear, theory-backed diagnostics.
- Model-agnostic and frequency-aware.
- Instance-level insight (predictable energy vs. error)
- Efficient: no extra training; few hyperparameters and straightforward implementation.

**Weaknesses:**

- Linear-only notion of predictability: The methodology defines “predictability” via linear coherence (SCP) and evaluates utilization via LUR. This assumes linear, LTI-style dependencies capture what matters. The paper does not establish when this proxy is sufficient. A good way to do this is probably demonstrate tightness of the linear lower bound against a strong linear forecaster, there has been many paper showing this in the past ([1](https://arxiv.org/abs/2205.13504)), ([2](https://arxiv.org/abs/2411.02796)), ([3](https://openreview.net/pdf?id=wfyc8vLcq0)).

- Equal-length constraint (history = horizon): The framework and experiments fix \(|x|=|y|=N\) to share a DFT grid. This is atypical in practice (commonly \(N_h \neq H\)) and may bias frequency resolution/weighting. Sensitivity or an extension to unequal lengths is missing.

- Boundary correction handles only mean jump: SCP subtracts a mean mismatch at the split but not slope/curvature changes. If a trend shift occurs at the boundary, the bound can over-attribute error to “unpredictable” power.

- Univariate treatment for inherently multivariate problems: Analyses are largely per-channel.

- Limited model coverage in key figures: Central plots (e.g., per-sample “predictable energy vs. error”) highlight a single baseline (DLinear). Generality claims are weaker without parallel plots for stronger nonlinear models.

- No direct evidence that spectral diagnostics beat time-domain diagnostics: Frequency-domain analysis is motivated but not compared against time-domain counterparts.

**Questions:**

- Does the framework and its conclusions go beyond forecasting? Can it extend to anomaly detection or tasks that do not assume a strict past→future order (e.g., retrospective labeling, two-sided contexts)?

- Similar approach also work on time-domain right? is there any intuition or direct evidence that working on spectral domain is better than that?

---

> ### Author Response · Authors · 2025-11-22
> **Response to Reviewer dD4o (Part 1)**
>
> We thank the reviewer for the constructive feedback and for recognizing the theoretical clarity and practical strengths of our diagnostics. We respond to the specific concerns and questions as follows.
>
> **W1**: Linear-only notion of predictability
>
> Our notion of predictability is defined via linear coherence, and the associated MSE_lb is a lower bound for Bayes–optimal linear predictors. We show that this linear proxy is sufficient for our purposes in two ways: (i) **it is tight with respect to the optimal linear forecaster in Gaussian settings**, and (ii) **on real benchmarks, it remains well aligned with both strong linear baselines and modern nonlinear models**.
>
> First, we explicitly validate our predictability score $ \mathcal{P} $ and the theoretical error lower bound MSE_lb on a Gaussian process with additive noise at varying SNRs ($\underline{\text{Fig. 1}}$). For each SNR, we evaluate an optimal linear forecaster and report its test MSE, the estimated lower bound MSE_lb, and the corresponding $ \mathcal{P} $. As the noise level decreases (higher SNR), $ \mathcal{P} $ increases monotonically toward one, and the model MSE approaches MSE_lb. **Across all SNRs, MSE_lb remains strictly below the realized MSE, confirming it is a valid lower bound, and the gap becomes very small at high SNR.** This validates both the calibration (monotone response to noise) and the tightness of our estimates in the linear regime.
>
> On real-world datasets, the assumptions behind the bound are only approximate, so we do not treat MSE_lb as an exact universal lower bound. Instead, we use it as a linear baseline and diagnostic. To check that this proxy is still informative, we compare it against a strong linear forecaster, DLinear. Across datasets and horizons, **DLinear’s MSE is close to MSE_lb, and their per-sample values are highly correlated**, indicating that the coherence-based bound captures most of what a well-tuned linear model can achieve on these benchmarks. **At the same time, SCP remains strongly correlated with errors of nonlinear models (e.g., Transformer-based forecasters) in our experiments **($\underline{\text{see Table 1}}$)**.**
>
> **W2**: Relaxing the equal-length constraint (history $ \neq $ horizon)
>
> In our framework, we adopt the equal-length setting $ |x| = |y| = N $ in the core experiments for two reasons. First, changing the history length $ N_x $ alters the balance between useful signal and nuisance context, effectively giving some models an extra advantage by pruning distracting history. Second, the intrinsic predictability of the future given the past is itself a function of $ N_x $, so models would no longer be compared under a controlled information condition. For these reasons, we fix the look-back window to the prediction horizon (history = horizon) for all models and datasets, ensuring a matched-information setting for both the SCP/MSE_lb estimates and the model comparisons.
>
> However, to address the concern regarding practical robustness (where typically $ N_x \neq N_y $), we performed the requested sensitivity sweep (Horizon fixed to $ N_y = 336 $, varying History $ N_x \in \{96, 192, 336, 720\} $). The results are:
>
> | Model | Metric | 96 | 192 | 336 | 720 |
> | --- | --- | --- | --- | --- | --- |
> | iTransformer | MSE | 0.491 | 0.479 | 0.471 | 0.480 |
> | iTransformer | R | 0.808 | 0.842 | 0.899 | 0.832 |
> | DLinear | MSE | 0.491 | 0.480 | 0.447 | 0.449 |
> | DLinear | R | 0.788 | 0.840 | 0.872 | 0.853 |
> | Predictability | MSE_lb | 0.431 | 0.433 | 0.404 | 0.424 |
>
> As shown below, while absolute MSE varies with history length, the correlation between SCP and model error remains consistently high (typically $ R>0.80 $). **These results indicate that our SCP-based predictability measure is robust to reasonable changes in the history window, even when **$ N_x \ne N_y $**.**
>
> We have clarified this design choice and added the above robustness check in the revised $\underline{\text{Appendix D.3}}$.
>
> **W3**: Boundary correction handles only mean jump
>
> For any model that only observes the history window $ x $, a future segment whose trend changes exactly at the split is effectively a local regime shift. Such behavior is intrinsically hard to predict from a short past window, and in our experiments ($\underline{\text{Fig. 3}}$), these regions indeed coincide with drops in SCP and sharp increases in forecasting error across models. In this sense, attributing the corresponding error to “unpredictable” power is appropriate: it reflects genuine task difficulty rather than a deficiency of the boundary correction.

---

> ### Author Response · Authors · 2025-11-22
> **Response to Reviewer dD4o (Part 2)**
>
> **W4**: Multivariate Extension
>
> We appreciate the suggestion to broaden the scope. In the revised version, we (i) **present SCP in a unified matrix form** that covers both univariate and multivariate cases $\underline{\text{(see Appendix C.1)}}$, and (ii) **add a controlled multivariate experiment** showing how SCP behaves as more informative input channels are revealed $\underline{\text{(see Appendix D.2)}}$.
>
> In the multichannel case with $ d_x $ input dimensions and $ d_y $ target dimensions, the residual cross-spectral density is defined as
>
> $ \widehat S_e(f)=\widehat S_{yy}(f)-\widehat S_{yx}(f)\bigl(\widehat S_{xx}(f)+\varepsilon I_{d_x}\bigr)^{-1}\widehat S_{xy}(f), $
> where $ \widehat S_{xx}(f)\in\mathbb{C}^{d_x\times d_x} $, $ \widehat S_{yy}(f)\in\mathbb{C}^{d_y\times d_y} $, and $ \widehat S_{xy}(f)\in\mathbb{C}^{d_x\times d_y} $.
>
> Aggregating over frequencies via the trace, $ \widehat{\mathrm{Var}}(\mathbf e)=\sum_{f\in\mathcal{F}}\mathrm{tr}\,\widehat S_{e}(f), \widehat{\mathcal{P}}^{\mathrm{multi}}=1-\frac{\widehat{\mathrm{Var}}(\mathbf e)}{\widehat{\mathrm{Var}}(\mathbf y)}$.
>
> We also construct $ d_x=6 $ sinusoidal input channels with distinct DFT-aligned frequencies and random phases, and one scalar target
> $ y(n)=\sum_{i=1}^{6} x_i(n)+\epsilon(n), \epsilon(n)\sim\mathcal{N}(0,0.05), $
> so that most target energy is linearly generated by the inputs. For each $ m\in\{1,\dots,6\} $, only $ (x_1,\dots,x_m) $ are revealed, averaged over $ N_{\text{samples}}=640 $ sequences. The results are:
>
> | $ m $ | $ \mathcal{P}^{\mathrm{multi}} $ ($ \mu\pm\sigma $) | $ \mathrm{MSE}^{\mathrm{multi}}_{\mathrm{lb}} $ ($ \mu\pm\sigma $) |
> | --- | --- | --- |
> | 1 | $ 0.117\pm0.121 $ | $ 0.486\pm0.067 $ |
> | 2 | $ 0.208\pm0.145 $ | $ 0.436\pm0.080 $ |
> | 3 | $ 0.360\pm0.130 $ | $ 0.353\pm0.072 $ |
> | 4 | $ 0.523\pm0.169 $ | $ 0.263\pm0.094 $ |
> | 5 | $ 0.662\pm0.151 $ | $ 0.186\pm0.083 $ |
> | 6 | $ 0.848\pm0.030 $ | $ 0.084\pm0.017 $ |
>
> As $ m $ increases, $ \mathcal{P}^{\mathrm{multi}}(m) $ increases in a monotonic fashion, while $ \mathrm{MSE}^{\mathrm{multi}}_{\mathrm{lb}}(m) $ decreases accordingly. These results show that **SCP extends naturally from scalar to multichannel settings and correctly tracks the gain in predictability from additional informative dimensions**.
>
> **W5**: Limited model coverage in figures
>
> We appreciate this suggestion. In the revised version ($\underline{\text{Fig. 3}}$), we have added parallel plots for a representative nonlinear model (PatchTST) alongside DLinear. These additional figures show that the qualitative behavior is consistent across architectures: model errors systematically vary with intrinsic predictability.
>
> **W6 & Q2:** Spectral diagnostics vs. time-domain diagnostics
>
> In principle, second-order time-domain and frequency-domain views are equivalent via the Wiener–Khinchin theorem: autocovariances and spectra encode the same information. Here, a simple controlled example illustrates where naive time-domain summaries can be misleading while coherence (and thus SCP) still captures linear predictability.
>
> Consider two noiseless signals $ x_t = \sin(\omega_0 t), y_t = \cos(\omega_0 t). $ Here $ y_t $ is a phase-shifted version of $ x_t $, obtained by a linear time-invariant transformation, so $ y $ is perfectly linearly predictable from $ x $. If we look only at the zero-lag Pearson correlation $\rho_{xy}(0) = \mathrm{corr}(x_t, y_t), $ and average over many periods (treating $ \theta = \omega_0 t $ as uniform on $ [0,2\pi] $), we get
>
> $ \mathbb{E}[\sin\theta\cos\theta] = 0, $so $ \rho_{xy}(0) = 0 $.
>
> **A simple time-domain diagnostic based solely on zero-lag correlation would thus suggest that **$ x $** and **$ y $** are “unrelated,” even though **$ y $** is deterministically generated from **$ x $** by a linear filter.**
>
> In the frequency domain, both $ x $ and $ y $ have all their energy concentrated at the same frequency $ \omega_0 $. Their cross-spectrum at $ \omega_0 $ differs only by a constant phase factor, so the squared coherence
>
> $\gamma^2(\omega_0)=\frac{|S_{xy}(\omega_0)|^2}{S_{xx}(\omega_0)\,S_{yy}(\omega_0)}=1$.
>
> In our framework, this implies a linear MSE lower bound of zero and SCP equal to one: the spectral diagnostic correctly recognizes that $ y $ is fully predictable from $ x $ despite the phase shift.
>
> Our use of the spectral domain is therefore motivated by practicality rather than a claim of strict dominance over time-domain analysis: (i) **spectral methods give an **$ O(N\log N) $** route**, whereas equivalent time-domain calculations (e.g., involving large covariance matrices or extensive lag sweeps) can be computationally heavier, often **scaling as** $ O(N^2) $; (ii) **LUR’s band-wise decomposition** is native in the frequency domain and directly exposes frequency-dependent failure modes; and (iii) the same machinery **extends cleanly to multivariate settings**. We have added a short discussion ($\underline{\text{see Appendix C.4}}$) to clarify.

---

> ### Author Response · Authors · 2025-11-22
> **Response to Reviewer dD4o (Part 3)**
>
> **Q1**: Does the framework go beyond forecasting (e.g., anomaly detection, two-sided contexts)?
>
> Yes, the framework extends naturally. The mathematical core requires only a paired input $ x $ and target $ y $ under a squared-error objective, allowing for flexible definitions:
>
> + **Anomaly Detection**: By sliding SCP over time, users can detect intrinsic anomalies. A sharp drop in SCP indicates that the "future" is no longer linearly explainable by the "past"—a signature of a structural break or regime shift. This distinguishes fundamental data shifts from mere model failures.
> + **Two-sided Contexts**: For this task, one can define $ x $ as the bidirectional context (surrounding window) and $ y $ as the masked center. In this setting, SCP quantifies the "Theoretical Restorability" of the missing segment, setting a baseline for imputation models.
>
> We sincerely hope our clarifications above have addressed your concerns and can improve your opinion of our work.

---

### Official Review · Reviewer_gxjx · 2025-10-31

**Soundness:** 3
**Presentation:** 4
**Contribution:** 3
**Rating:** 8
**Confidence:** 3

**Summary:**

The authors introduce two new metrics, Spectral Coherence Predictability (SCP), and. Linear Utilization Ratio (LUR). This aims to provide more thorough understanding of the model and data relationship and to provided directions to optimize/tackle in the model for time series forecasting.

**Strengths:**

- The authors propose metrics which aim to reduce guesswork when diagnosing model performances, which is a welcome notion in the community.
- Overall, the paper is well written.

**Weaknesses:**

--

**Questions:**

- How well do the authors believe their metrics hold for time-series extraction tasks, not forecasting? For example, in the rPPG domain. wherein the PPG time series is extracted from facial/skin video?

---

> ### Author Response · Authors · 2025-11-22
> **Response to Reviewer gxjx**
>
> We thank the reviewer for the positive assessment of our work, and respond to the specific question as follows.
>
> **Q1:** How well do the authors believe their metrics hold for time-series extraction tasks, not forecasting? For example, in the rPPG domain, wherein the PPG time series is extracted from facial/skin video?
>
> Conceptually, our framework only requires a pair of aligned time series $ (x, y) $ and a squared-error objective between them. In the main paper, we instantiate this as multi-horizon forecasting, where $ x $ is a history window and $ y $ is a future window of the same series, but mathematically $ y $ can be any target signal we aim to infer from $ x $, not necessarily the future.
>
> This makes time-series extraction tasks such as rPPG a natural fit. In an rPPG setting, **one can let **$ x $** be a temporal sequence of video-derived features and **$ y $** be the ground-truth PPG waveform over the same time window**. Under this pairing, our coherence-based construction yields:
>
> + an SCP-style quantity that measures what fraction of the PPG is linearly explainable from the video features, and
> + an LUR-style decomposition that shows which frequency bands (e.g., the narrow heart-rate band versus higher-frequency noise) are intrinsically well supported by the visual signal.
>
> Since PPG has a well-localized fundamental frequency and harmonics, such band-wise SCP/LUR analyses could be particularly informative in rPPG, indicating both the overall extractability of the signal and which components are inherently recoverable from the video.
>
> As discussed in our revised $\underline{\text{Appendix C.1}}$ (Multivariate Extension), the framework supports nonlinear feature maps. One could map raw video frames into a high-dimensional feature space first, then compute SCP to quantify predictability in that chosen representation, bypassing the limitations of raw pixel-space linearity.
>
> While our current empirical validation focuses on forecasting, the spectral coherence principles underpin signal extraction as well. We view the application of SCP/LUR to cross-modal tasks like rPPG as a promising direction for future work, potentially offering rigorous upper bounds on signal recovery.
>
> We sincerely hope our clarifications above have addressed your concerns and can improve your opinion of our work.

---

### Official Review · Reviewer_4paP · 2025-11-01

**Soundness:** 3
**Presentation:** 3
**Contribution:** 2
**Rating:** 4
**Confidence:** 3

**Summary:**

This paper investigates the issue of predictability in time-series forecasting, which aims to revolutionize the evaluation of time-series models. It proposes SCP for measuring difficulty of a given forecasting instance and LUR for measuring how effectively a model exploits the linearly predictable information within the data. Experiments are performed on public time-series datasets to verify the proposed claims

**Strengths:**

1. This paper investigates the evaluation of time-series forecasting, which is an interesting perspective in this field.

2. This paper provides a detailed algorithm table, which is helpful for readers to understand the flow of the algorithm.

3. The paper is written well with clear logic flows.

**Weaknesses:**

1. Authors note that `standard evaluation metrics conflate a model’s performance with the data’s intrinsic unpredictability`, which seems to be an important motivation of this work. However, to my view, this conflation does not impede the validity of existing metrics for evaluation. Notably, the data’s unpredictability seems to be fixed, so the rank of the standard metrics could reflect the rank of model’s performance despite the interference of data’s unpredictability.

2. Eq. 3 in preliminary seems to be $R^2$ ，which is a widespread metric; it would be not very necessary to introduce it with large volume of texts, a concise definition is sufficient.

3. The squared coherence in Eq.5 is an established metric. For example, it is included in the term `coherence (sometimes called magnitude-squared coherence)` in Wikipedia. However, no reference is provided but it is necessary to clarify the contribution of existing studies.

4. Rigorous theoretical evidence is necessary but lacking to demonstrate the benefits of the proposed metrics over the state-of-the-art standard evaluation metrics. The theoretical discussion in lines 211-215 is good, but it only concerns with the property of the proposed metric, without evidence on the advantage of the proposed metrics over the state-of-the-art standard evaluation metrics even on certain scenarios.

5. The proposed metrics could be linear, i.e., there would be no guarantee that they could be effective for non-linear datasets or models, which is a critical limitation in the deep learning age.

6. The dataset coverage is not sufficient. Currently there are only three types of datasets: ETT (the 4 subsets are often merged), ECL, Weather. As a evaluation-centered paper, diversity of datasets in terms of sample number, complexity, context, variable number, etc, should be critically considered in experiment design.

**Questions:**

Please see the weaknesses above

---

> ### Author Response · Authors · 2025-11-22
> **Response to Reviewer 4paP (Part 1)**
>
> We thank the reviewer for the thoughtful assessment and respond to the comments as follows.
>
> **W1 & W4**: On the role of standard error metrics and the need for predictability-aware evaluation
>
> Standard error metrics such as MSE remain valid and useful for ranking models. In fact, our predictability measure is explicitly aligned with MSE. Our aim is different: aggregate error alone does not disentangle model capacity from data difficulty, and therefore cannot support the kind of diagnostic analysis we aim to provide.
>
> First, the “intrinsic unpredictability” varies substantially (i) across datasets, (ii) across time within the same dataset (predictability drift), and (iii) across channels and frequency bands. **A single global MSE value cannot tell whether a high error comes from an intrinsically hard regime (low predictability) or from a model failing on an intrinsically easy regime (high predictability), as shown in Fig. 3**.
>
> Second, our theoretical discussion show that they are task-aligned and interpretable: in the linear setting, SCP is tightly linked to the Bayes-optimal MSE lower bound, and LUR decomposes this bound across frequency bands. **Standard error metrics do not offer such a link to Bayes risk or a band-wise decomposition.**
>
> Third, we provide empirical evidence that **SCP/LUR reveal the structure that MSE alone does not expose**. Standard metrics give only a single global ranking, whereas our goal is to understand how different models behave across predictability regimes and across frequencies. Our band-wise and predictability-aware analyses show clear differences between linear and nonlinear models, even when their overall MSE is similar.
>
> In summary, we view SCP/LUR as complementary to standard evaluation metrics: they are grounded in the same squared-error objective, provide a Bayes-risk interpretation, and offer additional diagnostic axes (predictability level and frequency band) beyond the usual error-based ranking.
>
> **W2**: Preliminaries (Eq. 3)
>
> Thanks for the suggestion. We have streamlined Eq. 3 and the surrounding explanation, reducing it to a concise Bayes-risk-based definition. Specifically, we now write:  ``Using the unconditional variance Var(y) as a baseline, we define intrinsic predictability as the normalized reduction of uncertainty``.
>
> **W3**: References for squared coherence (Eq. 5)
>
> We appreciate the reminder. We have added appropriate references.
>
> **W5**: Linearity of the proposed metrics and applicability to nonlinear data/models
>
> Our formal guarantees are indeed derived under linear assumptions, using linear prediction theory to tightly link SCP to the Bayes-optimal MSE lower bound in the linear setting. At the same time, our empirical results show that SCP remains well aligned with the errors of deep nonlinear forecasters across real-world benchmarks, indicating that it provides informative diagnostics beyond the idealized linear regime.
>
> First, the “linearity” in SCP is with respect to a chosen representation, not necessarily the raw observation space. In the revised version ($\underline{\text{see Appendix C.2}}$), we clarify that fixed nonlinear preprocessing can be applied before computing SCP: **one may map the original series into a higher-dimensional feature space via nonlinear transforms and then apply our framework to these transformed channels**. In this view, SCP measures how much of the future variance is linearly predictable from the history in the chosen feature space, which can capture a broad class of nonlinear relationships.
>
> Second, to support such high-dimensional feature spaces (and multivariate forecasting), we have formalized the multivariate extension in the revision. In the multichannel case with $ d_x $ input dimensions and $ d_y $ target dimensions, we define the residual cross-spectral density as
>
> $ \widehat S_e(f)=\widehat S_{yy}(f)-\widehat S_{yx}(f)\bigl(\widehat S_{xx}(f)+\varepsilon I_{d_x}\bigr)^{-1}\widehat S_{xy}(f), $
> where $ \widehat S_{xx}(f)\in\mathbb{C}^{d_x\times d_x} $, $ \widehat S_{yy}(f)\in\mathbb{C}^{d_y\times d_y} $, and $ \widehat S_{xy}(f)\in\mathbb{C}^{d_x\times d_y} $.
> We then aggregate over frequencies via the trace:
> $\widehat{\mathrm{Var}}(\mathbf e)=\sum_{f\in\mathcal{F}}\mathrm{tr}\,\widehat S_{e}(f), \widehat{\mathrm{Var}}(\mathbf y)
> =\sum_{f\in\mathcal{F}}\mathrm{tr}\,\widehat S_{yy}(f), $
> and define the multivariate SCP as
> $\widehat{\mathcal{P}}^{\mathrm{multi}}=1-\frac{\widehat{\mathrm{Var}}(\mathbf e)}{\widehat{\mathrm{Var}}(\mathbf y)}. $
>
> This formulation proves that SCP is mathematically ready to handle the multidimensional outputs produced by nonlinear feature maps, covering both scalar and multichannel settings. In the revised version, we add a dedicated discussion in $\underline{\text{Appendix C.1}}$.

---

> ### Author Response · Authors · 2025-11-22
> **Response to Reviewer 4paP (Part 2)**
>
> **W6**: Dataset coverage and diversity
>
> We agree that dataset diversity is important. In the original submission, we already followed the standard LTSF protocol and evaluated on widely used multivariate benchmarks: ETTh1/ETTh2, ETTm1/ETTm2, ECL, Weather, together with synthetic datasets.
>
> In the revised version, we further extend the experimental coverage by adding two additional benchmarks from different domains: **Traffic** (hourly road occupancy from 862 sensors in a large freeway network) and **ILI** (weekly influenza-like illness statistics, shorter and noisier with strong annual seasonality). The detailed results on these two datasets are summarized below.
>
> | Model | Metric | Traffic-96 | Traffic-192 | Traffic-336 | Traffic-720 | ILI-60 | ILI-72 |
> | --- | --- | --- | --- | --- | --- | --- | --- |
> | iTransformer | MSE | **0.394** | **0.385** | **0.388** | **0.416** | 2.001 | 2.186 |
> |  | MAE | **0.269** | 0.269 | 0.274 | 0.290 | 0.954 | 1.033 |
> |  | NMSE | 0.303 | 0.302 | 0.265 | 0.273 | 0.806 | 1.032 |
> |  | R | 0.849 | 0.917 | 0.959 | 0.968 | 0.687 | 0.847 |
> | TimeMixer | MSE | 0.485 | 0.423 | 0.407 | 0.437 | 2.272 | **1.928** |
> |  | MAE | 0.319 | 0.285 | 0.275 | 0.297 | 0.977 | **0.938** |
> |  | NMSE | 0.376 | 0.327 | 0.275 | 0.292 | 1.055 | 0.714 |
> |  | R | 0.919 | 0.939 | 0.971 | 0.960 | 0.783 | 0.781 |
> | DLinear | MSE | 0.649 | 0.459 | 0.436 | 0.450 | 2.671 | 2.661 |
> |  | MAE | 0.396 | 0.305 | 0.296 | 0.306 | 1.083 | 1.114 |
> |  | NMSE | 0.541 | 0.370 | 0.305 | 0.302 | 1.077 | 1.099 |
> |  | R | 0.899 | 0.929 | 0.970 | 0.968 | 0.850 | 0.919 |
> | PatchTST | MSE | 0.451 | 0.402 | 0.401 | 0.434 | **1.758** | 2.010 |
> |  | MAE | 0.288 | **0.263** | **0.267** | **0.289** | **0.863** | 0.948 |
> |  | NMSE | 0.356 | 0.321 | 0.277 | 0.288 | 0.753 | 0.792 |
> |  | R | 0.893 | 0.920 | 0.966 | 0.965 | 0.675 | 0.852 |
> | TimesNet | MSE | 0.606 | 0.608 | 0.630 | 0.672 | 2.160 | 1.994 |
> |  | MAE | 0.327 | 0.329 | 0.347 | 0.357 | 0.961 | 0.974 |
> |  | NMSE | 0.370 | 0.376 | 0.331 | 0.335 | 0.866 | 0.697 |
> |  | R | 0.960 | 0.969 | 0.946 | 0.983 | 0.820 | 0.742 |
> | Predictability | MSE_lb | 0.803 | 0.616 | 0.400 | 0.636 | 2.151 | 2.681 |
> |  | P | 0.514 | 0.619 | 0.760 | 0.610 | 0.560 | 0.466 |
>
> **Across both datasets, SCP and the linear MSE lower bound remain well aligned with the empirical errors.** On Traffic in particular, the correlation between per-sample MSE and SCP-based predictability is mostly above 0.9 across models and horizons.  The high correlation $ R $ confirms that SCP reliably diagnoses difficulty regardless of domain complexity or data scale.
>
> We have updated the dataset description and added an “Additional Dataset Evaluation” section in $\underline{\text{Appendix D.4}}$ to clarify this extended coverage. These benchmarks already span different sampling rates, scales, and numbers of variables.
>
> We sincerely hope our clarifications above have addressed your concerns and can improve your opinion of our work.

---

> > ### Comment · Reviewer_4paP · 2025-11-22
> >
> > Thank you very much for your detailed response.
> >
> > I also have preservation on the claim `standard evaluation metrics conflate a model's performance with the data's intrinsic unpredictability.`, which seems central to the motivation of this study. Could you please clarify in short (with 1-2 sentences) why it is necessary to disentangle model capacity from data difficulty for evaluation, especially (i) the latter could fixed given a dataset, and (ii) the theoretical benefits seem to be constrained for linear models. FYI, the second aspect could seem to be mean (for models) but it would be a necessary aspect to be considered theoretically in establishing an evaluation metric (which should be more theoretically grounded).
> >
> > Besides, `our theoretical discussion show that they are task-aligned and interpretable: in the linear setting, SCP is tightly linked to the Bayes-optimal MSE lower bound, and LUR decomposes this bound across frequency bands.` I also have preservation on the claim here. Bayes-optimal lower bound and decomposition across frequency bands do not necessarily imply task-alignment and interpretability.
> >
> > Therefore, I keep my rating conservatively at this stage.

---

> > > ### Author Response · Authors · 2025-11-25
> > > **Response to Reviewer 4paP**
> > >
> > > We thank the reviewer for raising these points, as they touch on the core motivation and scope of our framework. We clarify briefly below:
> > >
> > > **Q1: Why disentangle model capacity from data difficulty if the dataset is fixed?**
> > >
> > > **Even within a fixed dataset, predictability varies significantly across samples.** SCP provides an instance-level measure of task difficulty, helping us see which samples are well learned and which are not, and thereby informing model design and training strategies.
> > >
> > > **Q2: Is the theoretical benefit constrained for linear models?**
> > >
> > > **Although our guarantees are stated for linear predictors, the framework is both theoretically flexible and empirically robust.**
> > >
> > > * **Theoretical Extensibility:** The framework supports mapping observations into a high-dimensional feature space (via kernels or encoders). This extends the linear lower bound to capture nonlinear dependencies within that transformed space.
> > > * **Empirical Validity:**
> > >     * **(i) Dominant Linearity:** The marginal performance gap between linear baselines (e.g., DLinear) and SOTA nonlinear models (e.g., iTransformer) on these benchmarks suggests that the predictable signal energy is primarily linear.
> > >     * **(ii) Conservative Baseline:** In drifting data, complex nonlinear patterns are often unstable or noise-like. Excluding them from the definition of predictable energy serves as a conservative baseline, preventing the metric from overfitting to noise.
> > >     * **(iii) Local Approximation:** Forecasting models rely on short local windows. Within these restricted contexts, local stationarity (linear approximation) is often the dominant extractable structure.
> > >
> > > SCP strikes a crucial balance: it preserves the simplicity and interpretability of a linear formulation while empirically demonstrating high consistency with complex models across diverse datasets.
> > >
> > > **Q3: Why do Bayes-optimal bounds and spectral decomposition imply "Task-Alignment" and "Interpretability"?**
> > >
> > > - **Task-Alignment (Regression vs. Classification):**
> > > Prior predictability theories [1, 2] primarily target symbolic or discretized sequences, establishing an **upper bound on classification accuracy** (over symbolized or tolerance-based states). This fundamentally mismatches the regression nature of time-series forecasting. In contrast, our framework constructs a Bayes-optimal MSE lower bound. This is strictly task-aligned because it operates on continuous values and minimizes **the exact same objective function (MSE)** used in long-term forecasting.
> > >
> > > - **Interpretability (Direct Comparison):**
> > > Because our bound shares the same physical units as the loss function, it ensures interpretability. It allows for a direct comparison with a model’s realized MSE to quantify the performance gap.
> > >
> > > We will explicitly clarify this fundamental distinction between classification-based upper bounds and our regression-based lower bound in the revised manuscript.
> > >
> > > [1] *Limits of predictability in human mobility* (Science 2010)
> > >
> > > [2] *Quantifying and Estimating the Predictability Upper Bound of Univariate Numeric Time Series* (SIGKDD 2024)

---

### Official Review · Reviewer_umja · 2025-11-04

**Soundness:** 3
**Presentation:** 3
**Contribution:** 2
**Rating:** 4
**Confidence:** 4

**Summary:**

This paper argues that current forecasting metrics didn't distinguish the influence of model inadequacy and intrinsic data unpredictability. Based on this, they proposed a unified framework which contains Spectral Coherence Predictability (SCP) and Linear Utilization Ratio (LUR). This framework helps to show the predictability drift and an architectural trade-off.

**Strengths:**

1: This paper shows a wealth of visualizations including different types of figures and tables, which are easy to understand.

2: This paper clearly make irreducible error explicit from the model performance and address evaluation issues inside this.

3: Their new designed SCP and LUR offers insights into different diagnostics, showing the evidence of predicability shift.

**Weaknesses:**

1: From this paper's implementations, it seems that model's implementation depends on several factors like window length, type, ridge and second-order stationarity within windows. However, we didn't see enough sensitivity analysis or ablation study on these choices.

2: In this paper's methods and experiments, the current formulation seems all focuses on linear coherence. How about any multivariate scenarios? How about any conditional coherence scenarios? How's your framework extend to these scenarios? More explanations or discussions about this should be mentioned.

3: For SCP, it seems that it is derived under the assumptions that the related time series is locally stationary and it is linear. However, in real-world forecasting, the scenario is more like a nonlinear or nonstationary process. So these assumptions may not hold. When the signal contains abrupt shifts or external factors, SCP may overestimate or underestimate the real case. How your SCP adapt to the real-world forecasting?

4: For LUR, it seems that it has a strong dependence on frequency-band partitioning. LUR requires to divided into several frequency bands and then compute based on each band. Different band boundaries may lead to different LUR values. So, your results may reflect how the frequency bands were chosen rather than intrinsic model behavior.

**Questions:**

Most questions are listed in the Weakness.

Beside them, more questions: \
1: Can your SCP/LUR be adapted to irregular timestamps or missing segments?

2: How sensitive is SCP to the spectral estimation setup?

3: Is there a statistical metric or method to summarize LUR performance in more details?

---

> ### Author Response · Authors · 2025-11-22
> **Response to Reviewer umja (Part 1)**
>
> We thank the reviewer for recognizing the clarity and diagnostic value of our framework. We address your concerns below, supported by new experiments in the revision.
>
> **W1 & Q2**: Sensitivity to spectral estimation setup
>
> To address this, we expanded the sensitivity and uncertainty analysis of our framework with respect to the Welch hyperparameters used to compute SCP.
>
> First, we assess the sensitivity of SCP to Welch coherence hyperparameters with forecast horizon fixed to $ N=96 $. We vary three factors: the window-length fraction $ L_w/N $, the overlap ratio $ \rho $, and the tapering window (Hann, Hamming, Blackman). The configuration $ (L_w/N, \rho, \text{window}) = (0.25, 0.5, \text{Hann}) $ is used as the default, and one hyperparameter is varied at a time while the others are held fixed. A condensed summary over $ 10000 $ samples is:
>
> | Param | Val | SCP $ \mu\pm\sigma $ | MSE_lb $ \mu\pm\sigma $ |
> | --- | --- | --- | --- |
> | win-frac | 0.25 | $ 0.344\pm0.109 $ | $ 0.186\pm0.133 $ |
> |  | 0.30 | $ 0.345\pm0.110 $ | $ 0.186\pm0.133 $ |
> |  | 0.35 | $ 0.367\pm0.108 $ | $ 0.183\pm0.133 $ |
> | overlap | 0.45 | $ 0.362\pm0.109 $ | $ 0.184\pm0.134 $ |
> |  | 0.50 | $ 0.344\pm0.109 $ | $ 0.185\pm0.133 $ |
> |  | 0.55 | $ 0.351\pm0.109 $ | $ 0.185\pm0.135 $ |
> | window | Hann | $ 0.345\pm0.110 $ | $ 0.186\pm0.134 $ |
> |  | Hamm. | $ 0.351\pm0.110 $ | $ 0.185\pm0.133 $ |
> |  | Black. | $ 0.335\pm0.110 $ | $ 0.187\pm0.135 $ |
>
> **Across all configurations, both the mean and the standard deviation of SCP and the linear MSE lower bound change only marginally.** This indicates that SCP is robust to reasonable choices of Welch parameters, and that most of the variability is driven by the underlying nonstationarity of the data rather than by the estimator itself.
>
> Regarding the ridge term, we employ a fixed small value $ \varepsilon $ (on the order of $ 10^{-8} $) solely for numerical stabilization. It is not a tuning parameter and does not affect spectral properties for standard time series. We have added detailed sensitivity plots and confidence intervals in $\underline{\text{Appendix D.1}}$ to further support these findings.
>
> **W2**: Multivariate and conditional coherence scenarios
>
> We appreciate the suggestion to broaden the framework. In the revised version, we (i) **present SCP in a unified matrix form** that covers both univariate and multivariate cases $\underline{\text{(see Appendix C.1)}}$, and (ii) **add a controlled multivariate experiment** showing how SCP behaves as more informative input channels are revealed $\underline{\text{(see Appendix D.2)}}$.
>
> In the multichannel case with $ d_x $ input dimensions and $ d_y $ target dimensions, the residual cross-spectral density is defined as
>
> $\widehat S_e(f)=\widehat S_{yy}(f)-\widehat S_{yx}(f)\bigl(\widehat S_{xx}(f)+\varepsilon I_{d_x}\bigr)^{-1}\widehat S_{xy}(f),$
> where $\widehat S_{xx}(f)\in\mathbb{C}^{d_x\times d_x} $, $ \widehat S_{yy}(f)\in\mathbb{C}^{d_y\times d_y} $, and $ \widehat S_{xy}(f)\in\mathbb{C}^{d_x\times d_y}$.
>
>
> Aggregating over frequencies via the trace, $\widehat{\mathrm{Var}}(\mathbf e)=\sum_{f\in\mathcal{F}}\mathrm{tr}\,\widehat S_{e}(f), \widehat{\mathcal{P}}^{\mathrm{multi}}=1-\frac{\widehat{\mathrm{Var}}(\mathbf e)}{\widehat{\mathrm{Var}}(\mathbf y)},$
> yields a multivariate SCP that is consistent with the scalar definition. This unified construction is detailed in the revised appendix and also naturally applies after fixed nonlinear feature maps, by treating the transformed channels as inputs.
>
> To validate the multivariate behavior of SCP, we construct $ d_x=6 $ sinusoidal input channels with distinct DFT-aligned frequencies and random phases, and one scalar target
> $ y(n)=\sum_{i=1}^{6} x_i(n)+\epsilon(n),\epsilon(n)\sim\mathcal{N}(0,0.05), $
> so that most target energy is linearly generated by the inputs. For each $ m\in\{1,\dots,6\} $, only $ (x_1,\dots,x_m) $ are revealed when computing the multivariate SCP and MSE lower bound, averaged over $ N_{\text{samples}}=640 $ sequences. The results are:
>
> | $ m $ | $ \mathcal{P}^{\mathrm{multi}} $ ($ \mu\pm\sigma $) | $ \mathrm{MSE}^{\mathrm{multi}}_{\mathrm{lb}} $ ($ \mu\pm\sigma $) |
> | --- | --- | --- |
> | 1 | $ 0.117\pm0.121 $ | $ 0.486\pm0.067 $ |
> | 2 | $ 0.208\pm0.145 $ | $ 0.436\pm0.080 $ |
> | 3 | $ 0.360\pm0.130 $ | $ 0.353\pm0.072 $ |
> | 4 | $ 0.523\pm0.169 $ | $ 0.263\pm0.094 $ |
> | 5 | $ 0.662\pm0.151 $ | $ 0.186\pm0.083 $ |
> | 6 | $ 0.848\pm0.030 $ | $ 0.084\pm0.017 $ |
>
> As $ m $ increases,$ \mathcal{P}^{\mathrm{multi}}(m) $ rises approximately monotonically while $\mathrm{MSE}^{\mathrm{multi}}_{\mathrm{lb}}(m) $ decreases, reflecting that more target energy becomes linearly reconstructable from the visible inputs. These results show that **SCP extends naturally from scalar to multichannel settings and correctly tracks the gain in predictability from additional informative dimensions, accurately reflecting the underlying conditional dependency structure**.

---

> ### Author Response · Authors · 2025-11-22
> **Response to Reviewer umja (Part 2)**
>
> **W3**: Assumptions of local stationarity and linearity
>
> We acknowledge that SCP is formally derived from second-order statistics under local stationarity and linear prediction. In practice, we handle real-world complexity via local approximation:
>
> + **Non-stationarity and abrupt shifts:** SCP is computed using windowed (Welch) spectra, assuming only local stationarity within each look-back window. When abrupt regime changes occur, the local window captures the mismatch between history and future, resulting in a drop in SCP. This correctly tracks the "predictability drift" and aligns with the spikes in forecasting error observed in Fig. 3, rather than overestimating predictability.
> + **Non-linearity:** For nonlinear models, SCP serves as a meaningful linear predictor baseline. Empirically, we have validated that SCP maintains a high correlation with the prediction error of various nonlinear models (e.g., iTransformer). This indicates that SCP effectively captures the dominant difficulty factors even in nonlinear regimes, serving as a robust baseline.
> + **Nonlinear extensions:** The framework also supports nonlinear feature maps. One can first map raw observations into a feature space and then compute SCP on the transformed channels, so that the same linear lower-bound interpretation applies in that feature space. In the revised version, we add a dedicated discussion in $\underline{\text{Appendix C.2}}$.
>
> **W4 & Q3**: Dependence of LUR on band partitioning and summary metrics
>
> By construction, LUR is defined per frequency band, so its numerical values can change when band boundaries are modified. In the revised version, we explicitly vary the band partitioning used in the LUR analysis. The qualitative patterns across models remain unchanged: iTransformer consistently achieves higher LUR in low-frequency bands where most of the energy concentrates, while DLinear tends to perform better in higher-frequency bands. Thus, **the main conclusions of the paper are stable under reasonable choices of band partitioning.**
>
> To provide more compact statistical summaries of LUR, we additionally report an LUR-weighted “**frequency centroid**,” which captures whether the model is biased toward lower or higher frequencies. For example, we obtain centroids such as DLinear: $ 0.253 $, PatchTST: $ 0.241 $, and iTransformer: $ 0.239 $, indicating that DLinear places more emphasis on higher-frequency components than the nonlinear models. These scalar summaries complement the full band-wise plots and make it easier to compare models statistically.
>
> We have included the corresponding sensitivity plots $\underline{\text{(Fig. 4, 7, and 8)}}$ and metric definitions in $\underline{\text{Appendix D.1.2}}$.
>
> **Q1**: Adapted to irregular timestamps or missing segments?
>
> Our framework only requires a pair of aligned history–future segments on a (possibly resampled) time grid and does not depend on any particular forecasting architecture. For irregular timestamps, a standard approach in forecasting practice is to first map the data to a regular grid via interpolation or aggregation (e.g., hourly or daily bins) and then apply the same SCP/LUR computations on the resampled series. For missing segments, one can either impute short gaps (e.g., with simple interpolation) or restrict attention to the available history window, which effectively reduces the usable history length $ N_x $. In all of these cases, the SCP/LUR definitions remain unchanged; only the effective history segment fed into the spectral estimator is modified.
>
> To check robustness with respect to shorter or longer histories, we added an experiment on ETTh1 with prediction horizon fixed to $ N_y = 336 $ and history lengths $ N_x \in \{96, 192, 336\} $, using both iTransformer and DLinear. The results are:
>
> | Model | Metric | 96 | 192 | 336 |
> | --- | --- | --- | --- | --- |
> | iTransformer | MSE | 0.491 | 0.479 | 0.471 |
> | iTransformer | $ R $ | 0.808 | 0.842 | 0.899 |
> | DLinear | MSE | 0.491 | 0.480 | 0.447 |
> | DLinear | $ R $ | 0.788 | 0.840 | 0.872 |
> | Predictability | MSE_lb | 0.431 | 0.433 | 0.404 |
>
> **While absolute model error (MSE) fluctuates with history length, the correlation between MSE_lb and error remains consistently high (typically **$ R \geq 0.80 $** for both architectures). **In the revised version, we add a dedicated discussion in** **$\underline{\text{Appendix D.3}}$**.**
> This confirms that even if the "effective history" is truncated due to missing segments, SCP remains a robust and reliable diagnostic of the remaining predictability.
>
> We believe these clarifications and new results address the main concerns and can improve your opinion of our work.

---

> > ### Comment · Reviewer_umja · 2025-11-28
> >
> > Thanks for your detailed response. I think most of my previous concern have been largely addressed. In particular, I think your explanations on W1-W4 help to strengthen both the empirical analysis and clarity of your SCP framework. I will consider increasing my score a little bit. Thanks a lot.

---

> > > ### Author Response · Authors · 2025-12-01
> > >
> > > We sincerely appreciate your positive feedback and your decision to raise the score. We are glad to hear that our clarifications regarding sensitivity analysis, multivariate and nonlinear extensions, and stationarity assumptions (W1-W4) have satisfactorily addressed your concerns.
> > >
> > > Thanks again for your constructive feedback. These discussions and the additional experiments have significantly strengthened both the robustness and clarity of our framework. We have ensured that all these revisions are incorporated into the updated manuscript.

---

### Official Review · Reviewer_j2o6 · 2025-11-08

**Soundness:** 2
**Presentation:** 2
**Contribution:** 2
**Rating:** 2
**Confidence:** 4

**Summary:**

This paper proposes a new diagnostic framework for time series forecasting that separates model performance from the data’s inherent predictability. The authors introduce two key metrics: Spectral Coherence Predictability (SCP) and Linear Utilization Ratio (LUR). Experiments show that forecasting difficulty can drift over time (“predictability drift”) and that complex models perform better on less predictable data, while linear models excel on more predictable tasks.

**Strengths:**

1. The technical methodology is clearly presented, with detailed pseudocode provided. The released code demonstrates strong reproducibility.

2. The conclusion that complex models perform better on less predictable data, while linear models excel on more predictable tasks is insightful and provides valuable understanding of model behavior.

**Weaknesses:**

1. The scientific and engineering contributions of the current work are limited. The authors mainly apply two existing statistical methods to analyze six small-scale time series datasets and five baseline models. Such a limited experimental setup is insufficient to support the claimed conclusions. More comprehensive and convincing experiments are needed.

2. The paper lacks strong case study visualizations to illustrate how SCP and LUR analyze time series data in practice. Their characteristics are only reflected through numerical metrics, which contradicts the authors’ stated motivation of moving beyond simplistic aggregate scores.

3. The authors fix the historical look-back window to ensure experimental fairness. However, different models exhibit varying sensitivity to the look-back length, which is typically treated as a tunable hyperparameter in time series forecasting. It is recommended that the authors evaluate multiple look-back window lengths for the same prediction horizon to fully and fairly assess each model’s potential performance.

**Questions:**

1. The paper lacks case study visualizations and analyses of real forecasting scenarios. It remains unclear under what circumstances the annalysis of SCP and LUR actually helps models achieve better predictive performance.

2. The proposed SCP and LUR metrics should be more thoroughly discussed in relation to the traditional and widely used autocorrelation function (ACF) analysis in time series forecasting. The authors are encouraged to clarify both the differences and connections between these approaches.

3. The concluding discussion on model design is insufficiently developed. The authors should elaborate on what constitutes a “complex” model versus a “linear” model, and further analyze the Pareto frontier between these two categories for the given datasets.

---

> ### Author Response · Authors · 2025-11-22
> **Response to Reviewer j2o6 (Part 1)**
>
> We acknowledge the reviewer’s detailed feedback and positive comments on our methodology and conclusions. Before addressing the specific points, we would like to clarify the scope and contribution of our work. Our primary goal is not to engineer new forecasting architectures or optimize performance via hyperparameter tuning. Instead, we aim to introduce a task-aligned predictability and diagnostic framework for multi-horizon forecasting. We address your specific concerns below.
>
> **W1**: limited contribution and small-scale experiments
>
> We understand this perspective but view the scope of our work somewhat differently, and clarify both the conceptual contribution and the empirical coverage.
>
> First, while our construction builds on classical spectral tools, the contribution goes beyond simply applying existing methods: We **(i) derive a coherence-based bound** explicitly aligned with MSE, **(ii) formalize intrinsic predictability** as the Bayes risk, **(iii) design band-wise and predictability-stratified evaluation protocols** to expose model biases, and **(iv) quantitatively document predictability drift** over time and show how it interacts with model behavior.
>
> Second, regarding empirical coverage, the six real-world benchmarks we use are **standard benchmarks for Long-Term Time Series Forecasting**. The five forecasting models represent diverse inductive biases, **including classical linear/MLP-style models, convolutional temporal models, patch-based Transformers, and token-style Transformers**.
>
> Third, in response to this concern, we have further strengthened the empirical section in the revised version by: **(i) adding additional benchmarks such as Traffic and National Illness datasets**, which broaden the range of domains, dimensionalities, and sequence lengths $\underline{\text{(see Appendix D.4)}}$; **(ii) including a multivariate SCP extension** together with controlled examples that validate its behavior $\underline{\text{(see Appendix C.1 and D.2)}}$; and **(iii) adding an illustrative section showing how predictability can be used to shape training schemes**, including hard-sample mining and curriculum-style scheduling $\underline{\text{(see Appendix C.3)}}$.
>
> **W2**: Lack strong case study visualizations
>
> We agree that visual case studies are important to convey how SCP and LUR behave in practice. Our intent is precisely to go beyond single aggregate scores, and several figures in the paper already do this. **(i) Figs. 1–2 visualize how SCP summarizes task difficulty across datasets and channels**, and show that intrinsic predictability is strongly correlated with model performance. **(ii) Fig.3 explicitly depicts predictability drift over time**, highlighting that common forecasting benchmarks are far from stationary. **(iii) Our band-wise LUR and predictability-stratified evaluation plots provide frequency-wise and difficulty-wise views** that reveal systematic differences between models which are completely hidden by a single MSE/MAE number.
>
> We also note that other reviewers explicitly highlighted these diagnostics as a strength, for example describing the band-wise LUR views as providing “**actionable insights beyond aggregate MSE/MAE**” and emphasizing that SCP and LUR “**show evidence of predictability shift**”.
>
> In the revised version, we further enrich the case-study material by (i) refining the predictability-drift visualizations to overlay the per-horizon MSE trajectories of representative models $\underline{\text{(see Fig. 3)}}$, and (ii) adding a multivariate case study that visualizes how multivariate SCP increases as more informative input channels are revealed $\underline{\text{(see Fig. 9)}}$.
>
> **W3**: Evaluate multiple look-back window lengths
>
> To address this, we added an experimentwith prediction horizon $ N_y = 336 $ and history lengths $ N_x \in \{96, 192, 336, 720\} $, using both iTransformer and DLinear. The results are:
>
> | Model | Metric | 96 | 192 | 336 | 720 |
> | --- | --- | --- | --- | --- | --- |
> | iTransformer | MSE | 0.491 | 0.479 | 0.471 | 0.480 |
> | iTransformer | R | 0.808 | 0.842 | 0.899 | 0.832 |
> | DLinear | MSE | 0.491 | 0.480 | 0.447 | 0.449 |
> | DLinear | R | 0.788 | 0.840 | 0.872 | 0.853 |
> | Predictability | MSE_lb | 0.431 | 0.433 | 0.404 | 0.424 |
>
> The MSE of both models clearly varies with the history length, confirming that the choice of $ N_x $ does affect absolute forecasting accuracy. In contrast, **across all history lengths, the correlation **$ R $** between SCP and per-sample errors remains consistently high (always above 0.80 for both architectures)**. We have added a subsection $\underline{\text{in Appendix D.3 of the revised paper}}$.
>
> However, since the past determines inherent predictability, models can no longer be compared under controlled information conditions. Therefore, in the main text, we fix the look-back window to the prediction range (history = range).

---

> ### Author Response · Authors · 2025-11-22
> **Response to Reviewer j2o6 (Part 2)**
>
> **Q1**: How can SCP/LUR help improve prediction performance?
>
> In our view, SCP and LUR is diagnostic and design-oriented rather than to directly define a new forecasting architecture. Our empirical results show that **models behave very differently across predictability regimes (easy vs. hard samples) and frequency bands.** These patterns are directly actionable for model selection and design, for example via **hybrid architectures** (e.g., combining a low-frequency specialist with a high-frequency specialist) and **predictability-aware sampling schemes** (e.g., hard-sample mining or curricula that move from high-SCP to low-SCP segments).
>
> In the revised version, we make these connections more explicit by adding an appendix section $\underline{\text{(Appendix C.3)}}$ that sketches how SCP can be integrated into training. As a simple example, for a dataset with per-sample predictability
> $P^i \equiv P_{xy}(x^{(i)},y^{(i)}) $.
> We can reweight the loss as
> $L=\sum_{i=1}^M w^{(i)}\,\ell\big(\hat{y}^{(i)},y^{(i)}\big),
> w^{(i)} \propto \bigl(P^i\bigr)^{\alpha},$
> where the exponent $ \alpha $ controls whether training emphasizes high-SCP (intrinsically predictable, low-noise) segments or low-SCP (hard, noisy) segments. This illustrates a simple mechanism for curriculum-style scheduling or hard-sample mining driven by intrinsic predictability rather than ad-hoc heuristics.
>
> **Q2**: Relation to traditional ACF analysis
>
> The autocorrelation function (ACF) and the power spectral density (PSD)—which underpins our SCP—are mathematically equivalent via the Wiener–Khinchin theorem: they are Fourier duals and thus contain the same second-order statistical information. In this sense, our framework builds on the same underlying structure as traditional ACF analysis.
>
> While ACF is excellent for visually identifying seasonality and dependence decay in the time domain, our frequency-domain framework offers two advantages for diagnosing deep learning forecasters:
>
> + **Disentanglement of spectral bias.** Neural networks often exhibit frequency-dependent failure modes (e.g., fitting low-frequency trend but missing high-frequency texture). ACF mixes these components across lags, whereas LUR decomposes the explainable and unexplained energy band-wise, allowing us to pinpoint which frequency components a model systematically fails to capture.
> + **Predictive vs. descriptive focus.** Standard ACF describes self-similarity of a single series ($ y_t $ versus $ y_{t-k} $). SCP instead measures the cross-dependency between the entire history window $ x $ and future window $ y $, quantifying the fraction of future variance that is linearly explainable from the past under the MSE objective and yielding a lower bound on achievable MSE. This provides a task-aligned notion of intrinsic predictability that raw ACF plots do not directly supply.
>
> In summary, ACF remains a useful descriptive time-domain tool, while SCP/LUR provide complementary, MSE-aligned spectral diagnostics tailored to multi-horizon forecasting and model comparison. In the revised version, we add a dedicated discussion in $\underline{\text{Appendix C.4}}$ that makes these connections explicit.
>
> **Q3**: Complex vs linear models and the Pareto frontier
>
> We appreciate this opportunity to clarify the definitions and the complexity-performance trade-off. We define "Linear" models (e.g., DLinear) as those employing strict linear mappings from history to future. "Complex" models (e.g., TimesNet, iTransformer) refer to architectures utilizing non-linear mechanisms (convolutions, attention, mixing) with higher representational capacity. This distinction is about function class, not just model size.
>
> While a hardware-level Pareto sweep (Parameters/FLOPs vs. Error) is outside our scope, Figure 5 presents a predictability-stratified performance analysis. Instead of a global complexity frontier, we investigate how the "**effective frontier**" shifts across intrinsic predictability (SCP) regimes.  **Models are compared across bins of intrinsic predictability **$ \mathcal{P} $**, effectively examining how different architectures occupy the frontier in the (predictability, error) plane rather than in the (complexity, error) plane.**   This provides a nuanced design guideline: model choice should be conditioned on the intrinsic predictability (SCP) of the data, rather than treated as a single global ranking.
>
> A comprehensive Pareto analysis of forecasting error versus computational cost across a broader model family is valuable.
> However, it is orthogonal to our main objective of developing and validating a task-aligned predictability and diagnostic framework. we view such a comprehensive Pareto study as an interesting avenue for future work.
>
> We hope that the clarification of our contribution scope, along with the new case studies and variable-window experiments, has satisfactorily addressed your concerns. We respectfully ask you to reconsider your evaluation based on these revisions.

---

### Official Review · Reviewer_rdDk · 2025-11-09

**Soundness:** 3
**Presentation:** 3
**Contribution:** 3
**Rating:** 4
**Confidence:** 3

**Summary:**

The paper introduces a spectral-coherence–based evaluation framework for time-series forecasting that (i) estimates a linear MSE lower bound (SCP) aligned with the task loss and (ii) computes a Linear Utilization Ratio (LUR) to diagnose how much linearly explainable signal a model actually captures, with band-wise (frequency) breakdowns. Experiments on synthetic and real datasets show good calibration of the bound, reveal predictability drift over time, and uncover architecture–difficulty complementarity when metrics are stratified by predictability.

**Strengths:**

1. The evaluation framework is task-aligned and scalable: the lower bound matches the MSE objective and its time complexity is O(NlogN), enabling large-scale use.
2. The diagnostic perspective is sufficient. LUR with band-wise views provides actionable insights beyond aggregate MSE/MAE.
3. The empirical study is clear. Convincing synthetic and real-world evidence is provided , including per-bucket analyses that expose capability differences obscured by averages
4. In practical relevance, the framework can guide model selection, curriculum design, and evaluation protocols.

**Weaknesses:**

1. Assumption scope: Reliance on (local) stationarity and second-order structure; tightness under strong nonstationarity/nonlinearity/exogenous drivers is under-characterized.
2. Hyperparameter sensitivity: Limited analysis of Welch/windowing choices, band partitioning, and associated uncertainty.
3. Multivariate extension: Joint, multichannel formulations (partial coherence, coherence matrices) are left for future work.

**Questions:**

1. Can the method support adaptive windowing/change-point handling with uncertainty estimates?
2. Please report systematic sweeps over Welch/window parameters and provide bootstrap CIs for predictability and LUR.
3. On controlled processes, how do SCP/LUR correlate with MI/entropy-rate surrogates versus compute cost?

---

> ### Author Response · Authors · 2025-11-22
> **Response to Reviewer rdDk (Part 1)**
>
> We appreciate the reviewer's careful assessment and the positive feedback on the task alignment, scalability, and practical relevance of our framework for model selection, curriculum design, and evaluation. Below, we address the raised weaknesses and questions in turn.
>
> **W1:** Assumption scope: Limited characterization of tightness under strong nonstationarity/nonlinearity/exogenous drivers
>
> We appreciate the reviewer's clarification request regarding the scope of our assumptions. Our formal tightness guarantees are indeed derived under standard second-order conditions (local stationarity and linear prediction) and are intended to connect SCP to the Bayes-optimal MSE lower bound for linear forecasters.
>
> Empirically, **our real-world benchmarks are clearly nonstationary, with predictability varying over time** (see $\underline{\text{Fig. 3}}$), yet SCP remains well aligned with the observed performance of both linear and nonlinear models across horizons. When exogenous drivers are observed and included as covariates in the input–output pair, they are naturally handled within the same construction.
>
> Moreover, the construction is compatible with standard nonlinear preprocessing: one can **apply a nonlinear feature map to the time series and compute SCP in the transformed feature space**, so that nonlinear dependencies in the original variables become linear, while the same linear lower-bound interpretation holds for predictors operating on the transformed variables. We have added a brief discussion $\underline{\text{in Appendix C.2 of the revised paper}}$.
>
> **W2 & Q2**: Hyperparameter sensitivity: Welch window hyperparameters and band partitioning
>
> We thank the reviewer for this suggestion. We have expanded the sensitivity and uncertainty analysis of our framework along two axes: **(i) the Welch hyperparameters used to compute SCP**, and **(ii) the frequency-band partitioning used in the LUR analysis**.
>
> First, we assess the sensitivity of SCP to Welch coherence hyperparameters with forecast horizon fixed to $ N=96 $. We vary three factors: the window-length fraction $ L_w/N $, the overlap ratio $ \rho $, and the tapering window (Hann, Hamming, Blackman). We adopt $ (L_w/N, \rho, \text{window}) = (0.25, 0.5, \text{Hann}) $ as the default configuration and vary one hyperparameter at a time while keeping the others fixed.
> The summary over $ 10000 $ samples is:
>
> | Param | Val | SCP $ \mu\pm\sigma $ | MSE_lb $ \mu\pm\sigma $ |
> | --- | --- | --- | --- |
> | win-frac | 0.25 | $ 0.344\pm0.109 $ | $ 0.186\pm0.133 $ |
> |  | 0.30 | $ 0.345\pm0.110 $ | $ 0.186\pm0.133 $ |
> |  | 0.35 | $ 0.367\pm0.108 $ | $ 0.183\pm0.133 $ |
> | overlap | 0.45 | $ 0.362\pm0.109 $ | $ 0.184\pm0.134 $ |
> |  | 0.50 | $ 0.344\pm0.109 $ | $ 0.185\pm0.133 $ |
> |  | 0.55 | $ 0.351\pm0.109 $ | $ 0.185\pm0.135 $ |
> | window | Hann | $ 0.345\pm0.110 $ | $ 0.186\pm0.134 $ |
> |  | Hamm. | $ 0.351\pm0.110 $ | $ 0.185\pm0.133 $ |
> |  | Black. | $ 0.335\pm0.110 $ | $ 0.187\pm0.135 $ |
>
> **Across all configurations, both the mean and the standard deviation of SCP and the linear MSE lower bound change only marginally.** This indicates that SCP is robust to reasonable choices of Welch parameters, and that most of the variability is driven by the underlying nonstationarity of the data rather than by the estimator itself.
>
> Second, we vary the frequency-band partitioning used in the LUR analysis. As expected, the exact LUR values within a band shift when the band edges move, but **the qualitative patterns across models remain unchanged**: iTransformer consistently achieves higher LUR in low-frequency bands where most of the energy concentrates, while DLinear tends to perform better in higher-frequency bands. The band-wise conclusions are not an artifact of a particular partition choice.
>
> We also report the corresponding sensitivity plots and confidence intervals for SCP and band-wise LUR in $\underline{\text{Appendix D.1 of the revised paper}}$.

---

> ### Author Response · Authors · 2025-11-22
> **Response to Reviewer rdDk (Part 2)**
>
> **W3**：Multivariate extension
>
> We appreciate the suggestion to broaden the scope. In the revised version $\underline{\text{(see Appendix C.1 and D.2 of the revised paper)}}$, we (i) **present SCP in a unified matrix form** that covers both univariate and multivariate cases, and (ii) **add a multivariate experiment** showing how SCP behaves as more informative input channels are revealed.
>
> In the multichannel case with $ d_x $ input dimensions and $ d_y $ target dimensions, the residual cross-spectral density is defined as
>
> $\widehat S_e(f)=\widehat S_{yy}(f)-\widehat S_{yx}(f)\bigl(\widehat S_{xx}(f)+\varepsilon I_{d_x}\bigr)^{-1}\widehat S_{xy}(f),$
> where $\widehat S_{xx}(f)\in\mathbb{C}^{d_x\times d_x}$, $\widehat S_{yy}(f)\in\mathbb{C}^{d_y\times d_y} $, and $ \widehat S_{xy}(f)\in\mathbb{C}^{d_x\times d_y}$.
>
> Aggregating over frequencies via the trace,
>
> $\widehat{\mathrm{Var}}(\mathbf e)=\sum_{f\in\mathcal{F}}\mathrm{tr}\,\widehat S_{e}(f),\qquad \widehat{\mathcal{P}}^{\mathrm{multi}}
> =1-\frac{\widehat{\mathrm{Var}}(\mathbf e)}{\widehat{\mathrm{Var}}(\mathbf y)}$.
>
> To validate the multivariate behavior of SCP, we construct $ d_x=6 $ sinusoidal input channels with distinct DFT-aligned frequencies and random phases, and one scalar target
> $ y(n)=\sum_{i=1}^{6} x_i(n)+\epsilon(n),
> \epsilon(n)\sim\mathcal{N}(0,0.05), $
> so that most target energy is linearly generated by the inputs. For each $ m\in\{1,\dots,6\} $, only $ (x_1,\dots,x_m) $ are revealed, averaged over $ N_{\text{samples}}=640 $ sequences. The results are:
>
> | $ m $ | $ \mathcal{P}^{\mathrm{multi}} $ ($ \mu\pm\sigma $) | $ \mathrm{MSE}^{\mathrm{multi}}_{\mathrm{lb}} $ ($ \mu\pm\sigma $) |
> | --- | --- | --- |
> | 1 | $ 0.117\pm0.121 $ | $ 0.486\pm0.067 $ |
> | 2 | $ 0.208\pm0.145 $ | $ 0.436\pm0.080 $ |
> | 3 | $ 0.360\pm0.130 $ | $ 0.353\pm0.072 $ |
> | 4 | $ 0.523\pm0.169 $ | $ 0.263\pm0.094 $ |
> | 5 | $ 0.662\pm0.151 $ | $ 0.186\pm0.083 $ |
> | 6 | $ 0.848\pm0.030 $ | $ 0.084\pm0.017 $ |
>
> As $ m $ increases, $\mathcal{P}^{\mathrm{multi}}(m) $ increases in a monotonic fashion, while $ \mathrm{MSE}^{\mathrm{multi}}_{\mathrm{lb}}(m) $ decreases accordingly. These results show that **SCP extends naturally from scalar to multichannel settings and correctly tracks the gain in predictability from additional informative dimensions.** However, a full multivariate study on high-dimensional real-world datasets, which would require careful modelling of inter-variable dependency structure, is orthogonal to our current focus and we view it as follow-up work.
>
> **Q1**：Adaptive windowing/change-point handling with uncertainty estimates
>
> Yes. Conceptually, this is compatible with our approach: one can detect regime shifts, recompute power spectra and SCP/LUR within each segment, and then treat segments as approximately independent samples to obtain segment-level predictability together with uncertainty.
>
> In the present work, we focus on standard static forecasting models that apply the same mapping to all samples under a fixed training and evaluation protocol. Allowing the predictability estimator to adapt locally (via data-dependent windowing or segmentation) while keeping the models fixed would create a conceptual mismatch between the difficulty measure and the evaluated predictors. To keep comparisons clean and interpretable, we adopt a simple fixed-window protocol in the main experiments.
>
> **Q3**: On controlled processes, how do SCP/LUR correlate with MI/entropy-rate surrogates versus compute cost?
>
> On controlled linear–Gaussian processes, SCP has a direct connection to information-theoretic quantities: both past–future mutual information and our SCP/MSE_lb are governed by the same variance ratio $ \text{Var}(Y)/\mathrm{MSE}^\star $, where $ \mathrm{MSE}^\star $ is the Bayes-optimal (linear) prediction error. SCP is a monotone transform of this ratio, while MI/entropy-rate surrogates typically apply a logarithmic transform.
>
> Beyond the linear–Gaussian case, MI/entropy-rate estimators are substantially more expensive and less directly tied to the MSE objective used in multi-horizon forecasting. It targets a different notion of predictability: **an upper bound on next-step classification-style accuracy** (over symbolized or tolerance-based states). Estimators typically incur superlinear time complexity in the sequence length $N$ (often **between $ O(N^2) $ and $O(N^3)$** due to subsequence matching) and require large sample sizes to obtain statistically stable estimates.
>
> By contrast, our SCP (i) is explicitly task-aligned with multi-horizon forecasting under MSE, (ii) is computationally efficient in the sequence length (**overall cost** $ O(N\log N) $), and (iii) naturally extends to band-wise analysis.
>
> We thank you again for the constructive feedback, which has helped strengthen our paper. We hope these clarifications satisfactorily address your concerns.

---

### Author Response · Authors · 2025-12-01
**Summary of Revisions**

We sincerely thank all the reviewers for their insightful reviews and valuable comments, which have been highly instructive in further improving our paper.

**Summary of Contributions:**
In this work, we introduce the concept of "predictability" to the field of general time series forecasting. This metric characterizes task difficulty at the instance level and admits a frequency-domain decomposition, enabling fine-grained evaluation. Our framework **maintains theoretical simplicity while ensuring empirical consistency** across extensive experiments involving **both linear and nonlinear models on 8 real-world and 2 synthetic datasets**. We further demonstrate the phenomenon of "predictability drift" and reveal systematic model performance differences across samples of varying difficulty and across frequency bands.


**Positive Reception:**
We are encouraged by the positive reception from the reviewers, who highlighted the motivation as an "interesting perspective" and the method as "task-aligned and scalable," providing "actionable insights beyond aggregate MSE/MAE." They also commended the "clear empirical study," "strong reproducibility," and the quality of presentation, noting that the paper is "written well with clear logic flows" and features "a wealth of visualizations."

**Addressing Concerns:**
The reviewers also raised insightful and constructive concerns, which we have addressed as follows:

* **Sensitivity Analysis:** We provided a comprehensive sensitivity analysis of SCP with respect to spectral estimation parameters, and of LUR under different frequency band partitions, validating that our metrics are robust to reasonable parameter variations.
* **Multivariate Extension:** We extended the univariate SCP to a matrix-form multivariate SCP. Through experiments with varying numbers of variables, we verified that the framework extends naturally from scalar to multivariate settings and correctly tracks the gain in predictability from additional informative variables.
* **Nonlinear Extension:** We clarified that one can map raw observations into a feature space and then compute SCP on the transformed channels, so that the same linear lower-bound interpretation remains valid within that feature space.
* **Non-stationarity and Abrupt Shifts:** We emphasized that SCP assumes only local stationarity. When abrupt regime changes occur, the local window captures the mismatch between history and the future, resulting in a drop in SCP. This correctly tracks "predictability drift" and aligns with the spikes in forecasting error observed in Fig. 3, rather than overestimating predictability.

*Note: The resolutions to the above points have been explicitly acknowledged and affirmed by Reviewer umja.*

Furthermore, we incorporated additional datasets and experiments with varying look-back window lengths. We also provided comparisons with time-domain correlations and traditional symbolic sequence predictability, demonstrating our method's computational advantages and its strong alignment with the MSE objective.

**Conclusion:**
Despite the recent technical issues with OpenReview that may have limited further interactive discussion, we have made every effort to address all concerns by providing sufficient evidence and the requested results. Compared with the initial submission, $\underline{\text{the revised manuscript includes an additional 10 pages}}$, and all updates are highlighted in **blue**.

The valuable suggestions from the reviewers have been instrumental in shaping the paper into a more complete and rigorous form. We would be very happy to answer any further questions.

---

### Meta-Review · Area_Chair_xwPH · 2025-12-23

**Summary:**

This paper proposes a new evaluation/diagnostic framework for time series forecasting.

Strengths:
(1) task-aligned and scalable evaluation framework. (2) practical benefits for model selection, curriculum design, and evaluation protocols. (3) newly designed SCP and LUR offering diagnostics insights

Weaknesses:
(1) limited scope and assumption to stationarity and second-order structure, and univariate time series. (2) lack of hyperparameter studies. (3) limited experimental set-up (datasets, model family) to fully support the claims/conclusion. (4) validity of the underlying assumption of SCP/LUR. (5) unclear benefit of the proposed metrics over existing ones.

**Reviewer Concerns:**

Some concerns, including sensitivity, extension to multi-variate and nonlinearity, have been addressed.

**Reviewer Scores:**

It is unlikely that reviewers will change the scores.

---

### Decision · Program_Chairs · 2026-01-26

Reject